Applied and Environmental Science

# Experimentally Validated Reconstruction and Analysis of a Genome-Scale Metabolic Model of an Anaerobic Neocallimastigomycota Fungus

St. Elmo Wilken,[a] Jonathan M. Monk,[b] Patrick A. Leggieri,[a] Christopher E. Lawson,[c,d] Thomas S. Lankiewicz,[c,e] Susanna Seppälä,[a] Chris G. Daum,[f] Jerry Jenkins,[f,g] Anna M. Lipzen,[f] Stephen J. Mondo,[f] Kerrie W. Barry,[f] Igor V. Grigoriev,[f,h] John K. Henske,[a] Michael K. Theodorou,[i] Bernhard O. Palsson,[b,c] Linda R. Petzold,[j] Michelle A. O'Malley[a,c]

aDepartment of Chemical Engineering, University of California Santa Barbara, Santa Barbara, California, USA

bDepartment of Bioengineering, University of California San Diego, San Diego, California, USA

cJoint BioEnergy Institute, Lawrence Berkeley National Laboratory, Emeryville, California, USA

dBiological Systems and Engineering Division, Lawrence Berkeley National Laboratory, Berkeley, California, USA

eDepartment of Evolution Ecology and Marine Biology, University of California Santa Barbara, Santa Barbara, California, USA

fUS Department of Energy Joint Genome Institute, Lawrence Berkeley National Laboratory, Berkeley, California, USA

gHudsonAlpha Institute of Biotechnology, Huntsville, Alabama, USA

hDepartment of Plant and Microbial Biology, University of California Berkeley, Berkeley, California, USA

iAgriculture Centre for Sustainable Energy Systems, Animal Production, Department of Agriculture and the Environment, Harper Adams University, Newport, United Kingdom

jDepartment of Computer Science, University of California Santa Barbara, Santa Barbara, California, USA

**ABSTRACT** Anaerobic gut fungi in the phylum Neocallimastigomycota typically inhabit the digestive tracts of large mammalian herbivores, where they play an integral role in the decomposition of raw lignocellulose into its constitutive sugar monomers. However, quantitative tools to study their physiology are lacking, partially due to their complex and unresolved metabolism that includes the largely uncharacterized fungal hydrogenosome. Modern omics approaches combined with metabolic modeling can be used to establish an understanding of gut fungal metabolism and develop targeted engineering strategies to harness their degradation capabilities for lignocellulosic bioprocessing. Here, we introduce a high-quality genome of the anaerobic fungus *Neocallimastix lanati* from which we constructed the first genome-scale metabolic model of an anaerobic fungus. Relative to its size (200 Mbp, sequenced at $62\times$ depth), it is the least fragmented publicly available gut fungal genome to date. Of the 1,788 lignocellulolytic enzymes annotated in the genome, 585 are associated with the fungal cellulosome, underscoring the powerful lignocellulolytic potential of *N. lanati*. The genome-scale metabolic model captures the primary metabolism of *N. lanati* and accurately predicts experimentally validated substrate utilization requirements. Additionally, metabolic flux predictions are verified by $^{13}$C metabolic flux analysis, demonstrating that the model faithfully describes the underlying fungal metabolism. Furthermore, the model clarifies key aspects of the hydrogenosomal metabolism and can be used as a platform to quantitatively study these biotechnologically important yet poorly understood early-branching fungi.

**IMPORTANCE** Recent genomic analyses have revealed that anaerobic gut fungi possess both the largest number and highest diversity of lignocellulolytic enzymes of all sequenced fungi, explaining their ability to decompose lignocellulosic substrates, e.g., agricultural waste, into fermentable sugars. Despite their potential, the development of engineering methods for these organisms has been slow due to their complex life cycle, understudied metabolism, and challenging anaerobic culture requirements. Currently, there is no framework that can be used to combine

Address correspondence to Michelle A. O'Malley, momalley@ucsb.edu.

Anaerobic fungi thrive in the guts of herbivores where they chomp up plant material, but we know very little about them. Here, we present the first genome scale metabolic model to describe gut fungal metabolism, which will enable their engineering.

multi-omic data sets to understand their physiology. Here, we introduce a high-quality PacBio-sequenced genome of the anaerobic gut fungus *Neocallimastix lanati*. Beyond identifying a trove of lignocellulolytic enzymes, we use this genome to construct the first genome-scale metabolic model of an anaerobic gut fungus. The model is experimentally validated and sheds light on unresolved metabolic features common to gut fungi. Model-guided analysis will pave the way for deepening our understanding of anaerobic gut fungi and provides a systematic framework to guide strain engineering efforts of these organisms for biotechnological use.

**KEYWORDS** genome-scale metabolic model, $^{13}C$ metabolic flux analysis, nonmodel fungus, Neocallimastigomycota, flux balance analysis, *Neocallimastix lanati*, anaerobes, anaerobic fungi

Anaerobic gut fungi in the early-branching phylum Neocallimastigomycota are found in the digestive tracts of large mammalian herbivores, where they play an integral role in the lignocellulolytic microbiome of their host (1). These fungi have an unusual lifestyle involving both a vegetative state and motile zoospores, and contain mitochondrion-like hydrogenosomes that are similar to organelles observed in other anaerobic eukaryotes (2). Recent transcriptomic and genomic analyses revealed that these fungi harbor an incredible diversity of carbohydrate-active enzymes (CAZymes) that are tailored to excel at decomposing lignocellulosic plant biomass (3–7). Moreover, these fungi appear to organize their biomass-degrading enzymes into unique eukaryotic cellulosomes, which are large and dynamic extracellular complexes of enzymes that likely contribute to the efficiency of biomass degradation by substrate channeling and modularity (4). Given their ability to metabolize raw lignocellulose, anaerobic gut fungi are an appealing biotechnological platform to drive the conversion of lignocellulose into hydrolyzed sugars and, ultimately, into renewable chemicals via fermentation (8–10).

While the lignocellulolytic capabilities of anaerobic gut fungi motivate their biotechnological interest, they are temperature sensitive, anaerobic, relatively slow growing, and hindered by requirements for specialized media. Moreover, their genomes are extremely AT and repeat rich, which makes sequencing and genetic engineering challenging (7, 11, 12). No robust genetic engineering tools have been developed for this class of fungi, hampering classic molecular biology techniques that can be used to investigate, understand, and engineer their metabolism. Despite these challenges, experimental and omics data sets have emerged to elucidate some aspects of their metabolism (7, 13–16). However, a framework to combine these data in a systematic manner to understand and engineer anaerobic gut fungal metabolism for industrial applications is not available. In particular, there is no clear consensus on the pathways operating in their hydrogenosome, a mitochondrion-like organelle involved in their energy and hydrogen metabolism. Current hypotheses suggest that either an energetically unfavorable pathway involving pyruvate formate lyase is used to produce $H_2$ or a pathway involving pyruvate ferredoxin oxidoreductase that is not supported by extracellular metabolite measurements is used (14, 16).

Genome-scale metabolic models (GEMs) can be used to address these shortcomings, as they are well suited to act as knowledge base platforms for integrating multiomic data sets and have been successfully used to drive the engineering of both prokaryotes and eukaryotes (17–19). Moreover, by experimentally testing the predictions of a GEM, it is possible to systematically refine the understanding of the metabolism of an organism. This ability to probe the metabolism *in silico* is particularly appealing in the context of nonmodel microbes, such as the anaerobic gut fungi, where direct metabolic manipulation is challenging.

Here, we introduce a high-quality PacBio-sequenced genome (200 Mbp, 62× sequencing depth) of the anaerobic gut fungus *Neocallimastix lanati*. Comparative genomic analyses revealed that *N. lanati* is metabolically similar to the other sequenced Neocallimastigomycota, including the presence of many CAZymes (~1,788 CAZymes,

**TABLE 1** Summary of the features of the genome of *N. lanati*[a]

| Feature[b] | Value |
| --- | --- |
| Genome size (Mbp) | 200.97 |
| No. of contigs | 970 |
| Sequencing read coverage depth | 62.05× |
| No. of predicted genes | 27,677 |
| No. of CAZymes | 1,788 |
| No. of GH genes | 678 |
| No. of GH genes containing a fungal dockerin domain | 271 |
| No. of metabolic genes | 2,761 |
| No. of transporters | 1,754 |

[a]The full genome (raw sequencing data, assembly, predicted genes and annotations) is available at https://mycocosm.jgi.doe.gov/Neolan1/Neolan1.info.html.

[b]CAZyme, carbohydrate active enzyme; GH, glycoside hydrolase. Metabolic genes are defined as genes that have an enzyme commission (EC) number assigned to them. GH genes that have a fungal dockerin domain are likely present in cellulosomal complexes (4).

with 585 associated with the fungal cellulosome), suggesting that insights gained from understanding its genome may be generalizable to other species in the clade. Therefore, we used the genome of *N. lanati* to construct the first genome-scale metabolic model of an anaerobic gut fungus. This fungus is well suited to act as a model system to investigate the metabolism of anaerobic gut fungi because, unlike many of them, it grows relatively well in completely defined (M2) medium (13, 20), it is relatively fast growing among gut fungal strains ($\mu = 0.045 \pm 0.003$ h$^{-1}$ in M2 medium) (21–23), and it can be cryopreserved.

The 3-compartment (extracellular, cytosolic, and hydrogenosomal compartments) model introduced here, named iNlan20, is composed of 1,018 genes, 1,023 reactions, and 816 metabolites and models the primary metabolism of *N. lanati*. The model is stoichiometrically consistent as well as mass and charge balanced. Experimental, genomic, transcriptomic, and metabolic flux analysis data were used to build and validate the model, which recapitulates extracellular metabolite production rates and accurately models the observed fungal growth rate. Furthermore, the model refines and expands on previous hypotheses regarding the metabolism of the gut fungal hydrogenosome. Both the model and experimental data suggest that pyruvate formate lyase (PFL) is significantly more active than pyruvate ferredoxin oxidoreductase (PFO) in the hydrogenosome but that hydrogen formation can only occur via the latter pathway. Going forward, this fungus and its associated model can be used to guide efforts to further refine aspects of gut fungal metabolism that remain unclear and direct metabolic engineering strategies. Indeed, model-based analysis could be invaluable in designing stable consortia between anaerobic gut fungi and other industrially utilized organisms—something that has not yet been fully realized (21, 24, 25).

## RESULTS

**The genome of *N. lanati* is rich in carbohydrate-active enzymes and metabolically similar to other anaerobic gut fungi.** Given the large repeat-rich genomes inherent to anaerobic gut fungi (12), PacBio sequencing was used to obtain a high-quality genome of the isolate *N. lanati*; this isolate was sourced from a fecal pellet of a sheep obtained from an enclosure at the Santa Barbara Zoo (the Index Fungorum identification number is IF557810). While the genome of this fungus is large (Table 1), it is the second least fragmented of all 5 published gut fungal genomes (see Table S1 in the supplemental material). The *N. lanati* genome encodes a rich array of carbohydrate-active enzymes (CAZymes) in numbers similar to those reported from other gut fungal genomes (Table S1) (3, 4, 7). In total, 1,788 CAZymes were identified in the genome, of which, 1,253 were expressed in the transcriptome. Like other anaerobic gut fungi, *N. lanati* deploys both complexed (cellulosomes) and uncomplexed CAZymes through its rhizoidal network (Table 1; Fig. 1), both of which contribute to the decomposition of

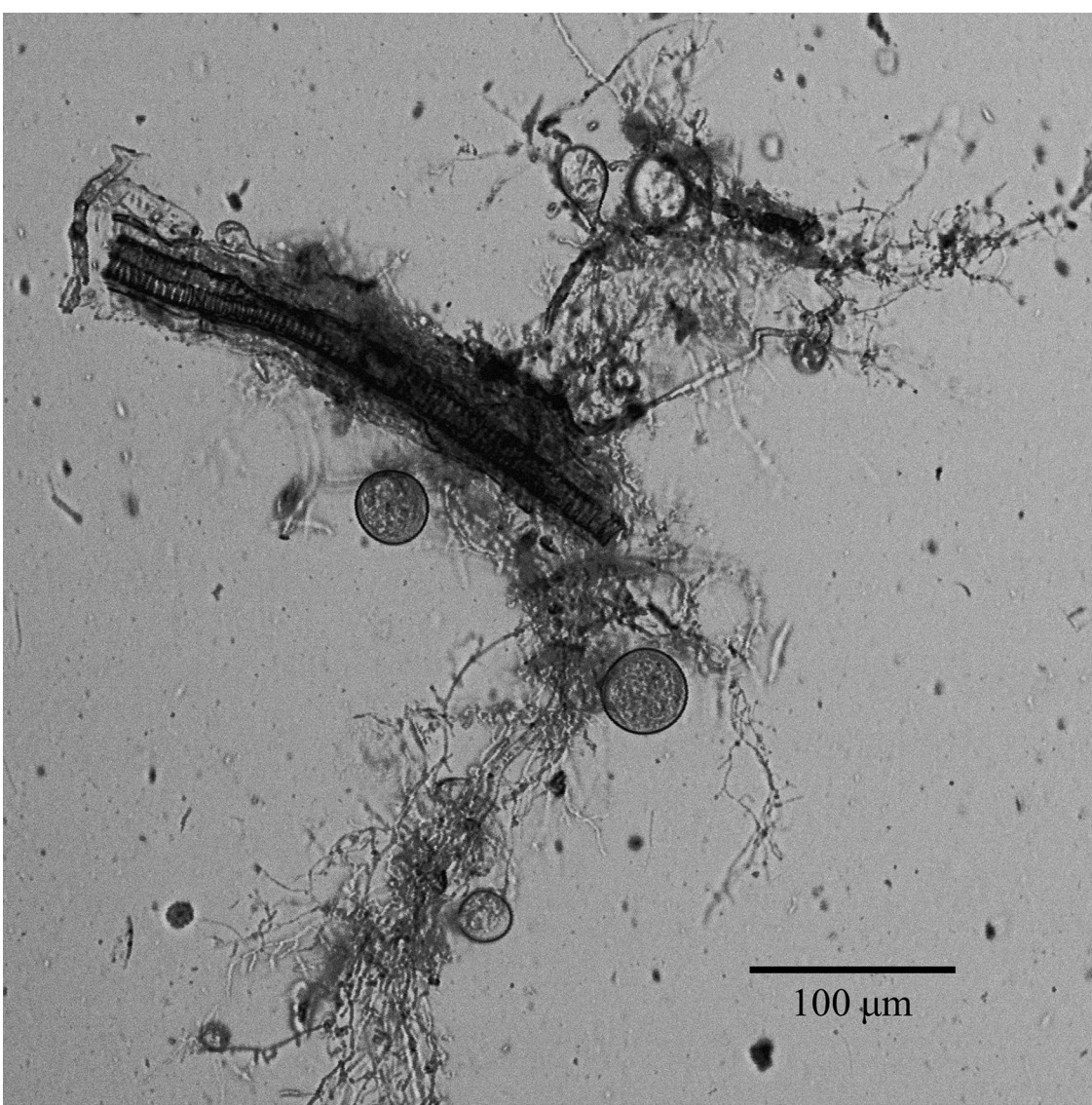

**FIG 1** The morphology of *Neocallimastix lanati* aids in the decomposition of unpretreated lignocellulose by disrupting the lignocellulosic plant biomass to increase the surface area available for enzymatic attack. A micrograph of a mature *N. lanati* sporangium growing on corn stover in M2 medium after 3 days of growth at 39°C. The filamentous rhizoidal network is used to increase the surface area for its lignocellulolytic enzymes that decompose the lignocellulosic corn stover into its fermentable sugar constituents.

lignocellulose. Figure S1 shows the breakdown of CAZyme domains identified in the genome of *N. lanati*. This, in combination with its relatively high growth rate on defined M2 media, suggests that *N. lanati* is a good model anaerobic gut fungus.

Despite advances in sequencing and annotation, a large number of putative gut fungal genes remain completely unannotated (~48% of the 27,677 predicted genes of *N. lanati*) (Table S1), which is consistent with previous genomic annotations in this clade. These unannotated genes contributed to gaps found in the draft reconstructed metabolism of *N. lanati*. A comparative genomic analysis within the primary metabolism across all high-quality publicly available gut fungal genomes (*Anaeromyces robustus*, *Neocallimastix californiae*, *Pecoramyces ruminatium*, and *Piromyces finnis*) revealed that the gut fungi are metabolically similar. Of the 1,023 unique EC numbers identified across these 5 genomes, fewer than ~3% are unique to each isolate (Fig. 2). This suggests that gut fungi share a similar primary metabolism. Thus, some metabolic gaps in

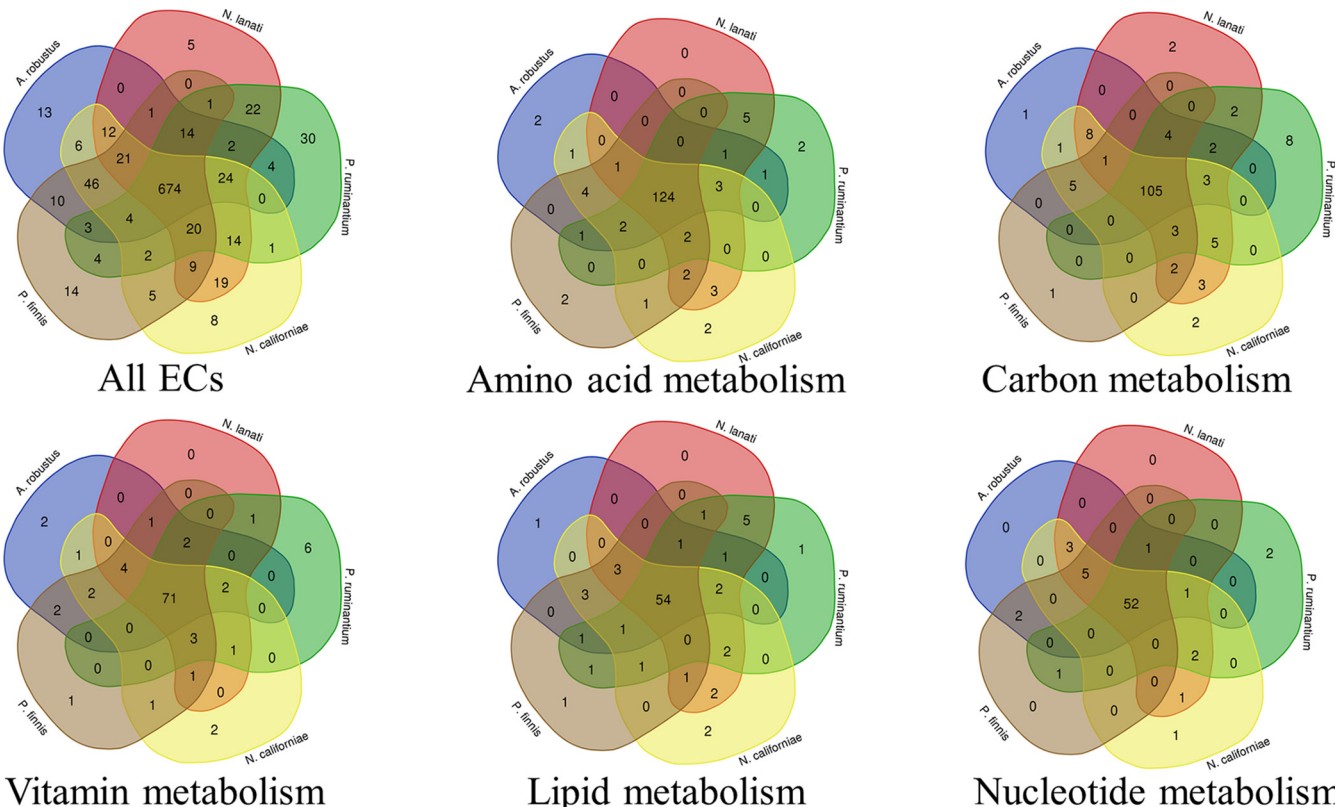

**FIG 2** Anaerobic gut fungi have very similar genetic metabolic potentials, suggesting that metabolic gaps can be filled by looking for homologous genes found in the other sequenced isolates. Each Venn diagram was generated by inspecting the intersection of the annotated EC numbers contained in the genome of each fungus for each metabolic module. Overlapping regions imply that those isolates share the EC assignments contained in each of the metabolic modules. The EC numbers contained in each module are based the KEGG database (60) (see supplemental data 4 in the iNlan20 GitHub repository available at https://github.com/stelmo/iNlan20 for the list of modules encompassing each Venn diagram), while the EC assignments for each fungus are based on the JGI and bidirectional annotation data as described in Materials and Methods.

the draft reconstruction could be filled by searching for genes in *N. lanati* that are homologous to those encoded, and annotated, in the genomes of the other gut fungal isolates. In this way, key enzymes in the biosynthesis pathways of arginine, asparagine, biotin, riboflavin, lipids, and fatty acids were identified and included in the metabolic reconstruction of *N. lanati*. In total, 35 gaps in the primary metabolic pathways were identified and annotated in this manner, as noted in the confidence score and homologous gene annotation fields in the model.

**The curated metabolic model of *N. lanati* captures the carbon, amino acid, vitamin, fatty acid, nucleotide, and lipid metabolism.** Based on the draft metabolic reconstruction of *N. lanati*, a manually curated genome-scale metabolic model of *N. lanati* was built (iNlan20) according to an established protocol for generating high-quality reconstructions (26). The model contains 1,023 reactions, 816 metabolites, and 1,018 genes distributed across 3 compartments (hydrogenosome, cytoplasm, and extracellular space). Where possible, experimental data were used to curate the model. Materials and Methods details specifics on the curation process, as well as experiments used to construct the biomass objective function of the model. Briefly, Table 2 shows the experimentally measured macromolecular components of *N. lanati* that were used to construct the biomass objective function for the genome-scale metabolic model. Further simplifying assumptions were made to construct the specific biomass objective function used in iNlan20. The insoluble carbohydrate component of the biomass was assumed to be solely chitin (27), and the amino acid composition of the protein component of the biomass was assumed to follow the amino acid distribution of the predicted genes (i.e., the predicted proteome). Similarly, the nucleotide composition was assumed to follow the composition of the genome (for the DNA nucleotides) and the

**TABLE 2** Experimentally measured macromolecular constituents of *N. lanati* that were used to construct the biomass objective function for the genome-scale metabolic model[a]

| Biomass component or function | Mass fraction (%) |
|---|---|
| Carbohydrate | 32.4 ± 1.6 |
| Protein | 43.7 ± 1.2 |
| Lipids | 4.9 ± 0.2 |
| DNA | 0.2 ± 0.1 |
| RNA | 0.6 ± 0.1 |
| Sum | 81.8 ± 3.2 |
| GAM | 76[b] |
| NGAM | 2.3[c] |

[a]Experimental data were used to estimate the biomass objective function. See Materials and Methods for more details.
[b]mmol ATP/g (dry weight)/h.
[c]mmol ATP/g (dry weight).

transcriptome (for the RNA nucleotides). The lipid component was assumed to be composed of myristic, palmitic, and stearic acids, which were found to be the major fatty acid components of the lipid fraction of *N. lanati* (see Fig. S2). The growth-associated and non-growth-associated maintenance (GAM and NGAM, respectively) functions were estimated using experimental data (see Fig. S3).

The model focuses on the primary metabolism but includes CAZymes as generalized cellulase and hemicellulase reactions. Of the 791 metabolic genes included in the model, 216 do not have gene assignments—reflecting the understudied nature of gut fungal metabolism. Despite this, the model is stoichiometrically consistent as well as mass and charged balanced (see Memote report or data set S1 in the iNlan20 GitHub repository available at https://github.com/stelmo/iNlan20). Table S2 explains the confidence rating assigned to the genes associated with each reaction in the model. In the energy-generating pathways, particular attention was paid to modeling the hydrogenosome (a mitochondrion-like organelle that functions completely anaerobically), which is discussed in greater detail in the following sections. More generally, the Embden-Meyerhof-Parnas variant of glycolysis is present in *N. lanati* as well as pathways for mixed-acid fermentation (succinate, acetate, lactate, formate, and ethanol), which are typically found in anaerobic gut bacteria (28). Interestingly, it was found that *N. lanati* possesses both the $NAD^+$ and $NADP^+$ variants of glyceraldehyde-3-phosphate dehydrogenase in glycolysis, with the latter used to conserve energy as NADPH instead of ATP. The pentose phosphate pathway of *N. lanati* is incomplete, with glucose-6-phosphate dehydrogenase and 6-phosphogluconate missing. These reactions regenerate NADPH and possibly explain the presence of the $NADP^+$ variant of glyceraldehyde-3-phosphate dehydrogenase as a compensating mechanism (29, 30). Despite these missing genes, the model is still capable of producing nucleotide precursors. The xylose isomerase pathway is also present in *N. lanati*, as has been found in other sequenced anaerobic gut fungi (21).

The major components (amino acids, nucleotides, vitamins, fatty acids, and lipids) of the anabolic metabolism of *N. lanati* were found to be present, in agreement with its ability to grow in M2 medium without these components added. Specifically, the complete biosynthesis pathways for all the proteogenic amino acids and the modeled fatty acids were found. Most of the canonical vitamin and cofactor (vitamin $B_5$, vitamin $B_6$, riboflavin, and thiamine) biosynthesis pathways were found to be complete, with the exception of folate, where no synthesis mechanism of 4-aminobenzoate was found. The heme and biotin biosynthesis pathways appeared to be incomplete; however, the latter vitamin was not found to be essential (experimentally) in defined medium, suggesting that the pathway is either poorly annotated or that the fungus does not require it for growth. Finally, the model recapitulates the experimentally observed

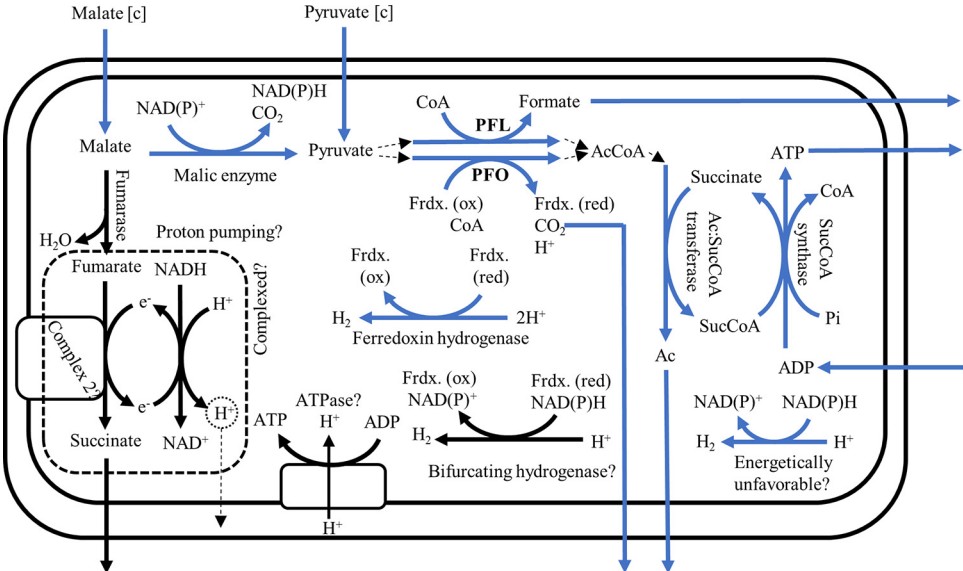

**FIG 3** An expanded model of the hydrogenosome is included in the model based on genomic annotation, literature, and predicted localization data (14–16). Core hydrogenosome enzymes are colored in blue, while speculative enzymes are shown in black. PFL, pyruvate formate lyase; PFO, pyruvate ferredoxin oxidoreductase; Ac, acetate; SucCoA, succinyl coenzyme A; CoA, coenzyme A; AcCoA, acetyl coenzyme A; Frdx, ferredoxin.

growth rate in defined medium using only the measured flux of glucose (1.5 mmol/g [dry weight]/h) as an input constraint (flux balance analysis predicted $\mu = 0.044$ h$^{-1}$ versus an experimentally measured $\mu = 0.045 \pm 0.003$ h$^{-1}$).

**iNlan20 includes an expanded model of the hydrogenosomal metabolism.** Anaerobic gut fungi possess a variant of the hydrogenosome, with the core set of enzymes that catalyze the conversion of malate and pyruvate to acetate, H$_2$ and formate already identified, as shown in Fig. 3 (2, 14, 16, 31). However, the metabolic pathway leading to H$_2$ production is not resolved, with literature suggesting either pyruvate ferredoxin oxidoreductase (PFO) or pyruvate formate lyase (PFL) as possible routes (Fig. 3). Both enzymes were identified in the genome and transcriptome and are thus included in the model of the hydrogenosome.

By combining literature sources, gene annotation, transcriptomic expression, and subcellular localization data, we have included additional pathways in the model of the hydrogenosome for *N. lanati* (Table 3 and Fig. 3). Of note is the inclusion of a putative ATP synthase (32, 33), which was previously speculated to be present in other anaerobic gut fungal isolates (7, 16). Additionally, we also found evidence that complex 2 of the mitochondrial electron transport chain is present: homologs to all four subunits were found to be expressed and localized to the hydrogenosome (complex 2, subunits A, B, C, and D). We could not find any homologs of the membrane-bound subunits of complex 1 or the ATP synthase in the *N. lanati* genome, as was also reported previously for other anaerobic gut fungi (34). A possible explanation for this is that many of the membrane-bound subunits of the electron transport chain are encoded by mitochondrial DNA, which the fungal hydrogenosome appears to have lost. Despite this, it is perplexing that no homologs of the membrane-bound subunits of complex 1 were found, since these are used to shuttle electrons between the two complexes in the inner membrane of the mitochondrion, and it remains unclear how complex 2 could function without them. However, homologs of the soluble subunits of complex 1, nuoF and nuoE, are highly expressed relative to the other core enzymes of the hydrogenosome (Table 3). The presence of the soluble subunits, coupled with the absence of the membrane-associated subunits of complex 1, has also been observed in the hydrogenosomes of, e.g., *Trichomonas vaginalis* (35, 36) and *Nyctotherus ovalis* (37).

**TABLE 3** Enzymes included in the model of the hydrogenosome metabolism

| Enzyme[a] | Gene (protein ID)[b] | Mean expression (TPM)[c] | Localization[d] | No. of other gut fungi where this gene was found |
|---|---|---|---|---|
| PFL 1 | 981064 | 1967 | Cytoplasm | 5 |
| PFL 2 | 1027775 | 182 | Cytoplasm | 5 |
| PFO | 623223 | 17 | Mitochondrion | 4[e] |
| Ac:SucCoA trans | 1731457 | 217 | Cytoplasm | 5 |
| Ac:SucCoA trans | 1316948 | 217 | Cytoplasm | 5 |
| SucCoA syn sub A | 1636158 | 1048 | Mitochondrion | 5 |
| SucCoA syn sub B | 1276456 | 1544 | Mitochondrion | 5 |
| Hydrogenase 1 | 1341048 | 219 | Mitochondrion | 5 |
| Hydrogenase 2 | 1718044 | 17 | Cytoplasm | 5 |
| Complex 1: nuoF | 1047445 | 339 | Mitochondrion | 5 |
| Complex 1: nuoE | 993995 | 519 | Mitochondrion | 5 |
| Complex 2: sub A | 1702000 | 4 | Mitochondrion | 5 |
| Complex 2: sub B | 1688149 | 13 | Mitochondrion | 5 |
| Complex 2: sub C | 1286787 | 12 | Mitochondrion | 3 |
| Complex 2: sub D | 1677752 | 8 | Mitochondrion | 2 |
| Fumarase | 985684 | 4 | Cytoplasm | 5 |
| ATP syn: sub alpha | 1037070 | 1 | Mitochondrion | 5 |
| ATP syn: sub beta | 1706307 | 8 | Mitochondrion | 5 |
| ATP syn: sub delta | 1045818 | 26 | Mitochondrion | 5 |
| ATP syn: sub gamma | 1061751 | 3 | Mitochondrion | 5 |

[a]PFL, pyruvate formate lyase; PFO, pyruvate ferredoxin oxidoreductase; Ac, acetate; SucCoA, succinyl coenzyme A; syn, synthase; trans, transferase; sub, subunit.
[b]ID, identifier.
[c]TPM, transcripts per million. Transcriptomic expression count data are derived from the M2 cellobiose expression data set and represent the means from triplicates for each enzyme.
[d]Localization was predicted using DeepLoc (55). Mitochondrial localization probably implies hydrogenosomal localization due to their evolutionary relationship (7).
[e]Not identified in the genome of N. californiae; however, a transcript with close homology to PFO was identified.

This raises two possibilities. First, that *N. lanati* possesses a proton-pumping mechanism, but the missing membrane-bound genes are unannotated. While the membrane-bound subunits of complex 1 and the ATP synthase appear to be missing, we do find preliminary evidence of a pH gradient inside the hydrogenosome (see Fig. S4). This provides some support for a mechanism, possibly involving complex 1, 2, and the ATP synthase, that makes use of this gradient to generate energy, as found in other hydrogenosomes (32), if these genes are present but unannotated. Second, it is possible that its hydrogenase associates with the identified nuoF-like subunit of complex 1 to form a bifurcating hydrogenase, as has been speculated to occur in the hydrogenosome of *T. vaginalis* (32, 35). Indeed, we find high-homology sequences in the *N. lanati* genome to all three of the bifurcating hydrogenase subunits that have been enzymatically characterized in *Thermotoga maritima* (38) (see data set S2 in the iNlan20 GitHub repository available at https://github.com/stelmo/iNlan20). In this case, the function of the membrane-bound subunits of complex 2 and the soluble subunits of the ATP synthase are unknown. In either case, further experimental work is needed to clarify these questions.

Taken together, our expanded model of the hydrogenosome includes the core enzymes previously reported in other gut fungal species as well as a speculative bifurcating hydrogenase, ATP synthase, and proton-pumping module composed of the complex 1 and complex 2 enzymes identified in the *N. lanati* genome (similar to what has been found in other $H_2$-producing mitochondria [32]). Given the speculative nature of the proton-pumping mechanism, the ATP synthase, and the bifurcating hydrogenase, these reactions are constrained to carry zero flux in the base case model. These constraints were modified to investigate the consequences of this extended hydrogenosomal metabolism, as discussed later. Additionally, it was previously suggested that a hydrogen dehydrogenase [$NAD(P)^+ + H_2 \leftrightarrow H^+ + NAD(P)H$] operates in the reverse

direction in the hydrogenosome (7, 14, 15). Consequently, this hydrogen dehydrogenase simultaneously produces $H_2$ and prevents the accumulation of NAD(P)H produced by the malic enzyme in the hydrogenosome. However, in this direction, the reaction is energetically very unfavorable, with a $\Delta G$ of $\approx 34 \pm 5.9$ kJ/mol assuming physiologically realistic conditions. Therefore, the flux bounds of this reaction in the hydrogenosome were set to reflect the assumption that the hydrogen dehydrogenase only carries flux in the forward, energetically favorable, direction.

**iNlan20 accurately predicts substrate utilization and *in vivo* fluxes.** The curated model was validated using a combination of growth curve, extracellular metabolite, and metabolic flux analysis (MFA) data. Substrate utilization tests were performed on 36 different carbon sources, focusing on metabolites that are present in the digestive tract environment of the anaerobic gut fungi (Table 4). The qualitative prediction accuracy of the model for the substrate utilization and vitamin essentiality validation tests is 89% (Matthews correlation coefficient, 0.79). Interestingly, despite the apparent presence of a full xylose isomerase pathway, *N. lanati* did not grow using xylose as its sole carbon source, as has been found in other gut fungi (21). In this case, the model's predictions were incorrect. It was previously suggested that transport limitations may cause this issue (21). However, a xylose transporter was identified in the genome, suggesting that cellular regulation might explain this discrepancy better. Vitamin essentiality tests were also conducted (Table 4). It was found that both heme and 4-aminobenzoate were essential for growth, in agreement with the model's predictions. In other gut fungi, heme has also been found to be essential (39), suggesting that its *de novo* biosynthesis pathway may be absent across the clade. It was found that only cysteine could be used as a sulfur source. However, it is not clear if this is a nutritional requirement, since every other reducing agent tested ($Na_2S$, 2-mercaptoethanol, and dithiothreitol) appeared to be toxic to the fungus. Since cysteine was used to ensure anaerobicity of the medium, we could not test nitrogen source utilization.

Metabolic flux analysis (MFA) was also used to experimentally verify the predicted intracellular fluxes of the GEM. A 1,2-$^{13}$C-labeled glucose tracer was used in conjunction with a carbon atom transition model built from the *N. lanati* metabolic reconstruction. For the MFA model, metabolic degeneracy caused by the ability of the hydrogenosome to metabolize both malate and pyruvate resulted in large bounds on the fluxes involving these metabolites. To circumvent this, the MFA model was constrained to only import pyruvate into the hydrogenosome, based on previous observations (14). Extracellular metabolic product measurements (ethanol, formate, $H_2$, acetate, succinate, and lactate) were also used to constrain the MFA model. This resulted in accurate internal metabolic flux measurements based on a statistically significant fit between measured and simulated proteinogenic amino acid labeling patterns (Fig. 4). These $^{13}$C measured fluxes were then compared to the fluxes predicted using the GEM with independently measured metabolite flux constraints (Table 5). Parsimonious flux-based analysis (pFBA) was then used to find unique flux predictions. Using these constraints, the coefficient of determination between the pFBA and MFA simulation was found to be 0.98 (linear regression fit $P < 0.01$) (see Fig. S5). This suggests that the constrained metabolic model accurately predicts the steady-state measured intracellular fluxes of *N. lanati*.

**The core hydrogenosome metabolism uses PFO to produce hydrogen, but PFL carries the most flux.** There remains uncertainty regarding the presence of pyruvate ferredoxin oxidoreductase (PFO) and its relative importance in the hydrogenosomal metabolism of anaerobic fungi. Earlier enzymatic characterization of hydrogenosomal proteins in Neocallimastigomycota suggested that PFO is the primary route for $H_2$ production through an associated ferredoxin hydrogenase, as found in the hydrogenosomes of other organisms (16, 31, 32, 40). However, more recent studies suggest that PFO is either absent or of only marginal importance in the gut fungal hydrogenosomal metabolism (14, 15). These later studies suggest that pyruvate formate lyase (PFL), which was likely acquired through horizontal gene transfer from bacteria (41), is significantly more active than PFO. It has been suggested that hydrogen evolution occurs through a hydrogen dehydrogenase working in an energetically infeasible reverse

**TABLE 4** Substrate utilization results suggest that the model accurately captures phenotypic behavior of *N. lanati*[a]

| Substrate | Growth[b] | |
| --- | --- | --- |
| | Model prediction | Exptl observation |
| Carbon utilization | | |
| Glucose | + | + |
| Cellobiose | + | + |
| Sorbitol | + | − |
| Fructose | + | + |
| Galactose | + | − |
| Maltose | + | + |
| Mannose | + | − |
| Sucrose | + | + |
| Xylose | + | − |
| Arabinose | − | − |
| Rhamnose | − | − |
| Pyruvate | − | − |
| Succinate | − | − |
| Citrate | − | − |
| Glycerol | − | − |
| Pectin | − | − |
| Cellulose | + | + |
| Lignocellulose | + | + |
| Acetate | − | − |
| Fumarate | − | − |
| N-Acetyl-glucosamine | − | − |
| Lactate | − | − |
| Maltodextrin | + | + |
| Methanol | − | − |
| Oxaloacetate | − | − |
| Xylan | + | + |
| Ethanol | − | − |
| Malate | − | − |
| Formate | − | − |
| Raffinose | + | + |
| Phenylalanine | − | − |
| Arginine | − | − |
| Leucine | − | − |
| Proline | − | − |
| Serine | − | − |
| Threonine | − | − |
| Vitamin essentiality | | |
| Pyridoxine | + | + |
| p-Aminobenzoic acid | − | − |
| Biotin | − | + |
| Cyanocobalamin | + | + |
| Riboflavin | + | + |
| Folic acid | + | + |
| Pantothenate | + | + |
| Nicotinic acid | + | + |
| Thiamin | + | + |
| Heme | − | − |

[a]The model accurately predicts phenotypic responses in 89% of the tested cases (Matthews correlation coefficient, 0.79). See Materials and Methods for details about the experiments that yielded these results.
[b]+, the model predicted growth/there was experimentally observed growth; −, no predicted growth/no experimentally observed growth.

direction (7, 14). Both PFO and PFL were identified in all published gut fungal genomes as well as in *N. lanati* (Table 3). The model was used to reconcile the role and relative importance of these two enzymes to hydrogenosome function under steady-state growth conditions.

Due to the reaction stoichiometry of PFL, the molar ratio of formate to acetate and

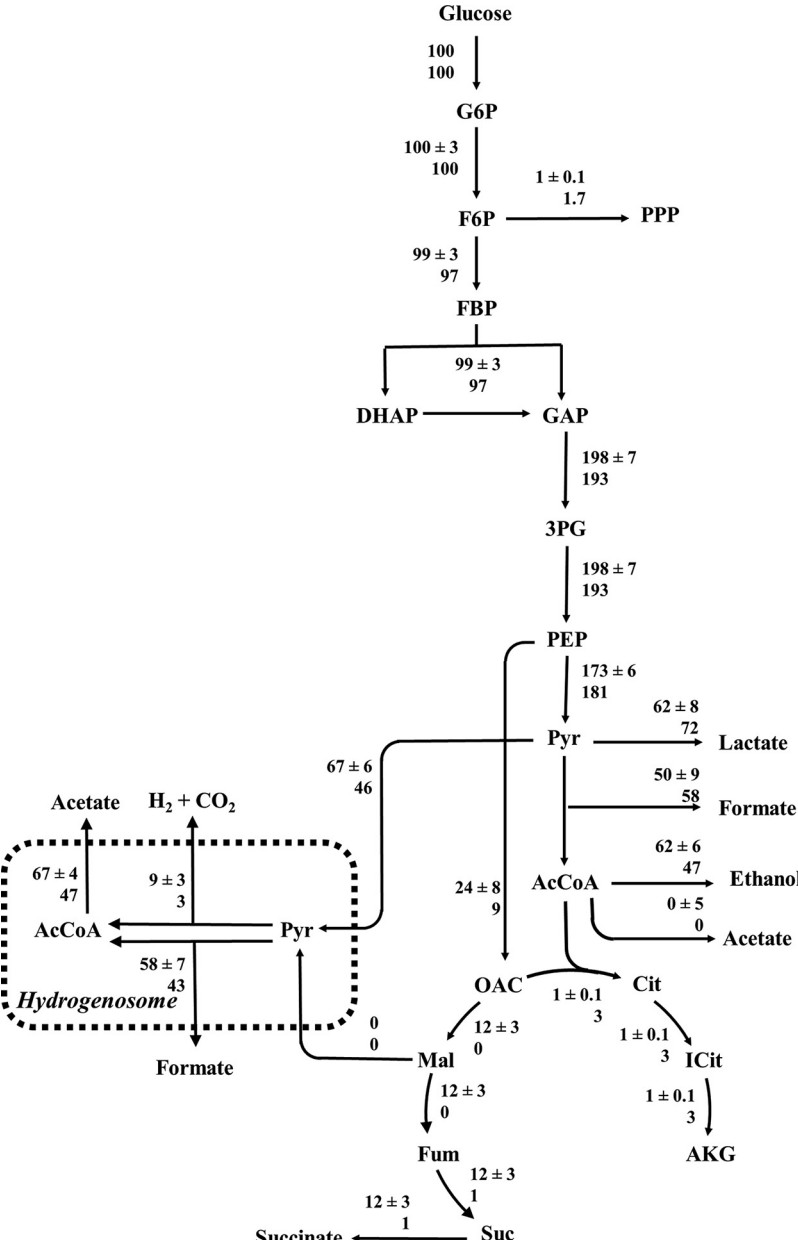

**FIG 4** The genome-scale metabolic model accurately predicts the *in vivo* carbon metabolism of *N. lanati*. Experimentally determined MFA fluxes and predicted pFBA fluxes (top and bottom, respectively) for glycolysis, the TCA cycle, and the hydrogenosome of *N. lanati*. Error estimates denote one standard deviation from the reported mean for the MFA measurements. Three serially passaged [1,2-$^{13}$C]glucose tracer experiments, grown in M2 medium at 39°C and harvested during exponential phase, were used to measure the *in vivo* fluxes (see Materials and Methods for more details and the model).

ethanol produced is expected to approach unity (1:1) if PFL is metabolically dominant. This is because ethanol and acetate are only produced from PFL (via acetyl coenzyme A) in the cytosol (15). Since PFO only produces acetate, and not formate, the ratio of formate to acetate and ethanol will not be unity if PFO carries significant metabolic flux. Figure S6A shows that the experimentally measured molar ratios of formate to acetate and ethanol are not significantly different (using the unequal variance test, $P <$ 0.05) from unity for *N. lanati*. This is in agreement with earlier metabolite measurements for *Piromyces* sp. strain E2, suggesting that PFL is dominant (14). Figure S6B shows that the unconstrained model predicts a wide range of possible ratios, reflecting

**TABLE 5** Experimentally measured external fluxes of various metabolites produced by *N. lanati* growing on cellobiose in M2 medium during exponential phase[a]

| Metabolite | Flux (mmol/g [dry weight]/h) | | | |
|---|---|---|---|---|
| | Mean | SD | Lower bound | Upper bound |
| Succinate | 0.03 | 0.01 | 0.02 | 0.05 |
| Lactate | 0.87 | 0.14 | 0.72 | 1.09 |
| Ethanol | 0.66 | 0.20 | 0.47 | 1.01 |
| Formate | 1.40 | 0.30 | 1.09 | 1.79 |
| Acetate | 0.56 | 0.12 | 0.42 | 0.71 |
| $H_2$ | 0.10 | 0.06 | 0.05 | 0.19 |

[a]See Materials and Methods for details.

the metabolic degeneracy of the carbon metabolism of *N. lanati*. Since there is no energetic cost associated with using PFO versus PFL (both produce one ATP molecule per pyruvate) (Fig. 3), the model predicts that both could be used to maximize ATP production in the hydrogenosome. However, external metabolite flux measurements show only modest $H_2$ production (Table 5), suggesting that cellular regulation may play a role in diverting flux to PFL instead of PFO. This can also be seen in the relative expression difference between PFL and PFO (an order of magnitude difference between them) in Table 3. When the model is constrained by the measured metabolite fluxes shown in Table 5, the range of possible ratios is reduced to those observed experimentally (Fig. S6B). Since PFO is the only (known) energetically feasible way to produce $H_2$, this result is not surprising. Using this constraint, the model suggests that PFL carries the most flux in the hydrogenosome, but that PFO is used to produce $H_2$.

**Electron bifurcation and proton pumping may form part of the hydrogenosomal metabolism.** Electron bifurcation is an energy conservation mechanism that can be used to drive thermodynamically unfavorable reactions by coupling endergonic and exergonic reactions through an enzyme complex (42). In the simplest case, this phenomenon is used by anaerobes to increase the yield of ATP through their carbon metabolism by using $H_2$ as an electron sink for the recycling of NADH to $NAD^+$ (42). In the case of the anaerobic gut fungi, the hydrogenosome can be used to generate 2 extra moles of ATP for every mole of glucose that enters glycolysis. However, not all the glycolytic flux can be diverted to the hydrogenosome, because $NAD^+$ needs to be regenerated from the NADH that is produced by glycolysis to maintain cellular redox balance. As mentioned before, $NAD^+$ is unlikely to be produced by the hydrogen dehydrogenase since the redox potential of NADH/$NAD^+$ is too electropositive to reduce $H^+$ directly (32). On the other hand, the ferredoxin-based hydrogenase included in the model only recycles the oxidized ferredoxin, produced by PFO, to reduced ferredoxin and does not impact the NADH/$NAD^+$ pools in the cell. Sequence homology suggests that the *N. lanati* hydrogenosome could potentially house a bifurcating hydrogenase, which would couple the reduction of $H^+$ to the oxidation of NADH through the ferredoxin produced by PFO. This hydrogenase enzyme complex would allow more flux to be channeled into the hydrogenosome for energy production, since the hydrogenase would now generate $NAD^+$ as well as $H_2$ (Fig. 3).

These findings suggest that there is a significant energetic advantage associated with possessing a bifurcating hydrogenase. The model captures this benefit by predicting a 16% increase in growth rate associated with the use of the bifurcating hydrogenase, as opposed to the ferredoxin hydrogenase ($\mu = 0.051$ h$^{-1}$ versus $\mu = 0.044$ h$^{-1}$, respectively). The model also dictates that the production of $NAD^+$ shifts from the cytosol to the hydrogenosome when the bifurcating hydrogenase is used, as shown in Fig. 5. However, this requires metabolic flux to be diverted from PFL to PFO in the hydrogenosome. Consequently, significantly more $H_2$ is predicted to be produced, which is not observed experimentally (Table S3). This discrepancy could be due to metabolic regulation that is unaccounted for in the GEM. Further experimental work needs

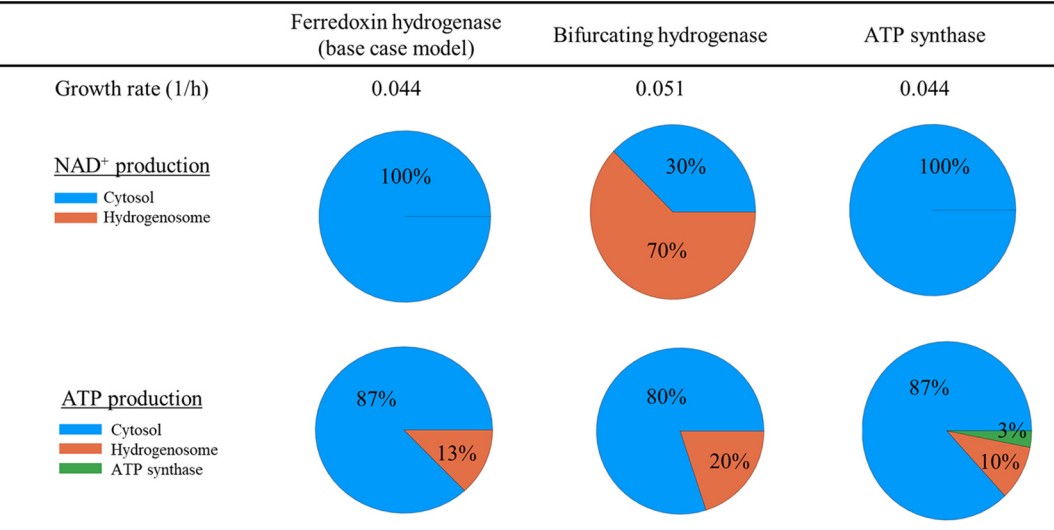

**FIG 5** The effects of including additional reactions in the hydrogenosome on $NAD^+$ and ATP production show that the putative bifurcating hydrogenase has a large positive effect on $NAD^+$ and ATP generation. Conversely, the putative ATP synthase has a negligible effect on both. The increase in growth rate caused by the putative bifurcating hydrogenase is due to $NAD^+$ being regenerated in the hydrogenosome; this allows more flux to be channeled into the organelle, which in turn produces more ATP. Flux sampling was used to determine the fluxes associated with $NAD^+$ and ATP production in each metabolic configuration. The base case model only includes the ferredoxin hydrogenase. The ferredoxin hydrogenase was replaced by a bifurcating hydrogenase to analyze its effect on the model. Finally, a complex 1, 2, and ATP synthase module was added to the base case model to investigate the consequences of this expanded metabolism. The model was constrained to produce biomass at 90% of the maximum yield; subsequently, 2,000 samples were drawn from each case. The average production of each metabolite in the hydrogenosome and cytosol are shown.

to be conducted to investigate the potential presence of the bifurcating hydrogenase in the anaerobic gut fungal hydrogenosome.

Given that the hydrogenosome in anaerobic fungi is relatively understudied yet related to the mitochondrion (32, 43), we assumed that their metabolite trafficking machinery is similar. Specifically, we assumed that the hydrogenosome has an inner membrane that is impermeable to $H^+$, like those of mitochondria (44). This implies that $H^+$ can only enter and leave the hydrogenosome through the action of transporters. The $H^+$ balance in the hydrogenosome has a direct effect on the ability of the putative ATP synthase to produce ATP. However, the impact of the ATP synthase on ATP production was found to be small, with it only supporting small fluxes ($\sim$3% of the glucose flux into the model) (Fig. 5). This suggests that the putative hydrogenosomal proton gradient (Fig. S4) may not be important for the generation of ATP, as is also suggested by the low expression of the ATP synthase complex subunits (Table 3), or that the proton gradient mechanism is not yet fully understood. Without further experimental evidence, it remains an open question whether the identified ATP synthase components are localized to the hydrogenosome or to some other subcellular organelle (34). Furthermore, the mechanism by which the putative complex 2 operates without the membrane-bound subunits of complex 1 remains to be determined.

**Metabolic degeneracy is related to the redox balance.** As is typical of unconstrained GEMs, the modeled gut fungal metabolism displays significant degeneracy, as shown by the high degree of flux variability (Table S4). The degeneracy is primarily due to the ability of *N. lanati* to regulate how $NAD^+$ is regenerated through its mixed acid fermentation pathways, i.e., through a combination of lactate dehydrogenase, acetaldehyde dehydrogenase, and alcohol dehydrogenase. Interestingly, the relative mean error between the predicted flux distributions and experimental measurements of the fermentation products is much more sensitive to constraints placed on acetate production than any other single measured external metabolite flux, as shown in Fig. 6.

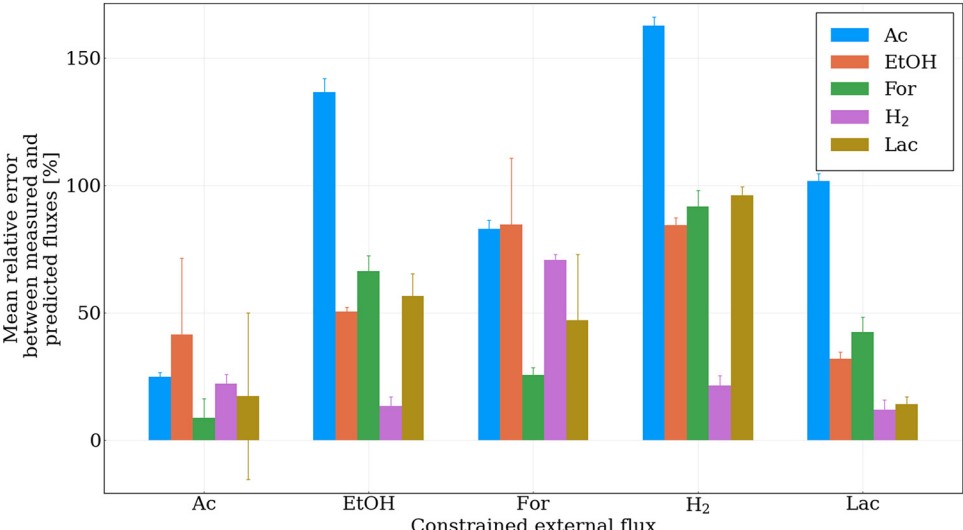

**FIG 6** The absolute relative error between the model predictions and the experimentally measured values suggest that constraining the flux of acetate production has the biggest impact on the model's accuracy. The flux of acetate (Ac), ethanol (EtOH), formate (For), $H_2$, and lactate (Lac) was constrained, individually, to their observed ranges (variables on the *x* axis). The resultant predicted fluxes of these metabolites (generated by sampling 2,000 possible solutions where the biomass objective function was within 90% of its optimal value and subject to the respective additional constraints as shown in the figure) were then compared to the experimental observations as shown in the legend.

Likewise, constraints placed on lactic acid flux also narrow the deviation of the predicted flux distributions. This effect is due to the different yields of $NAD^+$ that can be achieved per mole of pyruvate depending on which mixed-acid fermentation pathway, or combination thereof, is constrained. Without the bifurcating hydrogenase, $H_2$ production does not significantly impact the overall redox balance of the cell. This possibly explains why its constraint has the smallest effect on the flux variability predicted by the model and may allow the cell to fine tune its metabolism to suit the environmental needs, e.g., sugar availability, by up- or downregulating the flux channeled to the hydrogenosome (2).

## DISCUSSION

While anaerobic gut fungi specialize in lignocellulose decomposition, they are not as well understood as model microbes such as *Escherichia coli* and *Saccharomyces cerevisiae* (45). The most recent GEMs of *E. coli* and *S. cerevisiae* map 1,515 and 1,150 genes to reactions, respectively (46, 47). In comparison, iNlan20 maps 567 metabolic genes to reactions, mainly within the primary metabolism, with the balance being CAZymes. The large number of unannotated predicted genes (~13,300) (Table S1) likely form part of the fungal secondary metabolism (48). Moreover, of the remaining ~14,400 genes annotated with at least a single domain, only 2,761 were associated with EC numbers (Table S1), highlighting how undercharacterized this early-branching clade of fungi is. In iNlan20, annotation gaps are centered around transporters and lipid metabolism. With improvements in gene annotation and more experimental characterization, this coverage is likely to increase. Various simplifications were also made during the model construction that may affect the predictions. First, it was assumed that the carbohydrate content of *N. lanati* is wholly composed of chitin; this is a simplification, as gut fungi make use of other carbohydrate moieties as well (e.g., glycosylation sugars [12, 27]). Second, the biomass function only accounts for fatty acid synthesis and no other lipids, resulting in the bulk of the blocked reactions and dead-end metabolites being associated with lipid metabolism. While cell wall and membrane biosynthesis are clearly essential building blocks in cellular homeostasis, these metabolites were not included in the

objective function of the model because the gut fungal lipid composition is not well studied. Thus, expanding knowledge of lipid and cell wall anabolism will likely improve the model as well as possibly contributing to a deeper understanding of how the gut fungi secrete very large proteins (e.g., the cellulosome, ~1 MDa [4]).

Compared to the MFA estimates, the model-based flux predictions were the most accurate around the central glycolytic pathways (Fig. 4). The largest relative discrepancies were observed in the tricarboxylic acid (TCA) cycle, specifically in the reductive branch leading to succinate production: $12 \pm 3$ versus 1 (normalized flux units) for MFA and pFBA, respectively. In eukaryotes, the TCA cycle is typically involved in producing a proton gradient within the mitochondrion, which in the context of the gut fungi, would suggest that it may be localized to the hydrogenosome. Model-based analysis indicated that the proton-pumping module and associated ATP synthase were of negligible importance (Fig. 5). Yet, the experimental data were more inconclusive: while the expression of these genes was comparatively low (Table 3), the JC-1 staining suggested that the hydrogenosome did possess an electrochemical gradient (Fig. S4). The absence of any identified membrane-bound subunits of complex 1 and the ATP synthase makes it hard to reconcile these data. These ambiguous observations suggest that experimental characterization of the gut fungal hydrogenosome (e.g., through enzyme purification and proteomics) is critically important for improving the model and answering lingering questions surrounding the metabolic role of the hydrogenosome.

Under the growth conditions analyzed here, neither the model predictions nor the experimental data suggested that PFO carries high flux. Consequently, low $H_2$ production was expected and observed. However, literature sources indicate that stable gut fungal/$H_2$-consuming methanogen cocultures are readily formed synthetically and observed in nature (25, 49, 50). Given the tight association between $H_2$-consuming methanogens and gut fungi, it is surprising that the $H_2$ flux is low and not more tightly coupled to the central metabolism. It is possible that there may be more complex cellular regulation at play within the hydrogenosome regarding PFO and PFL than modeled here. Interestingly, there is both a large size difference between PFO and PFL (~539 kDa versus 90 kDa, respectively) and a catalytic rate difference (~$48 \pm 38$ s$^{-1}$ versus $8 \pm 5$ s$^{-1}$, respectively; data derived from BRENDA [51]). It is possible that under high sugar availability conditions, as used here, *N. lanati* produces the smaller slower enzyme (PFL) without compromising growth. In contrast, under more challenging conditions, e.g., when the fungus is using lignocellulose as a carbon source, it might produce more PFO. Alternatively, recent high-resolution growth rate data suggest that gut fungi have highly variable growth rates that fluctuate with environmental conditions (22). Thus, it is also possible that the gut fungi modulate the expression of PFO and, consequently $H_2$ production during their life cycle. High-temporal-resolution proteomics and transcriptomics that examine various life stages may shed more light on this question.

Despite these discussed caveats, the iNlan20 shows good agreement with experimental measurements. Due to their nonmodel nature, anaerobic gut fungi are vastly understudied, which presents unique challenges when constructing a metabolic model of these cryptic organisms. Model-based analysis represents a systematic framework that can be used to identify high-impact knowledge gaps and focus experimental attention, and in this case, the model indicates that focus should be directed toward a detailed characterization of the gut fungal hydrogenosome. Within the realm of nonmodel microbes, metabolic modeling is an appealing technique that can be used to speed up the biotechnological translation of anaerobic gut fungi and other nonmodel microbes.

**Conclusion.** Here, we have introduced a high-quality genome and transcriptome of a novel anaerobic gut fungus, *N. lanati*. While the genome is large, it is relatively unfragmented compared to the genomes of the other sequenced anaerobic gut fungi. Additionally, the genome encodes a large number and diversity of CAZymes, most of which are expressed in the transcriptome. This genome was used to construct the first genome-scale metabolic model of an anaerobic gut fungus. The model, iNlan20, accurately recapitulates the observed growth rate, *in vivo* fluxes, and substrate

consumption and requirement profiles. The model refines and expands on our understanding of gut fungal hydrogenosomal metabolism. We confirm previous findings that suggested that PFL carries more flux than PFO in the hydrogenosome, but an energetically favorable route to hydrogen production still requires the action of PFO. The possible presence of a bifurcating hydrogenase and/or a proton-pumping mechanism suggests that anaerobic fungi may have evolved more complex energy conservation mechanisms that allow them to compete with faster-growing rumen bacteria. Experimental work, likely involving the isolation, purification, and enzymatic characterization (through assays and proteomic analysis) of the hydrogenosome, is necessary to further refine our understanding of its metabolism. This model is well poised to serve as a platform to build a better understanding of these nonmodel organisms. Moreover, the model will serve as a valuable tool to systematically guide future engineering efforts of gut fungi for converting lignocellulose into value-added products.

## MATERIALS AND METHODS

**Metabolic reconstruction, visualization, and simulation.** All publicly available annotated genomes within the clade Neocallimastigomycota were downloaded from the Joint Genome Institute's (JGI) MycoCosm database (48). This includes the high-quality PacBio-sequenced genomes of *Anaeromyces robustus*, *Piromyces finnis*, and *Neocallimastix californiae* (4) as well as the novel isolate *Neocallimastix lanati* introduced here. The genomes of *Pecoramyces ruminatium*, also known as *Orpinomyces* sp. strain C1A (7, 52), and *Piromyces* sp. strain E2 (4) were also included for completeness. The gene annotation data supplied by the JGI was combined with annotations derived from bidirectionally searching by blast (using BLASTp [53]) the predicted genes from the gut fungal genomes against the curated Swiss-Prot database from UniProt (54). Briefly, bidirectional blast searching annotates a predicted gut fungal gene if (i) the top hit using the fungal genome as the query and the reference collection as the database is the same as when (ii) the gut fungal genome is used as the database and the reference collection is used as the query. Furthermore, only matches with E values smaller than $1e^{-20}$ were considered for assigning Enzyme Commission (EC) annotations to genes. This information was collated into a master metabolic table (see data set S3 in the iNlan20 GitHub repository available at https://github.com/stelmo/iNlan20) and subsequently used to construct the model and assign genes to reactions. Enzyme complexes were assigned by using the "Subunit structure" field in the UniProt database. Protein localization was predicted using DeepLoc (55). Reaction directions were primarily inferred from MetaCyc (56), and specific Gibbs free energy change of reactions reported were calculated using eQuilibrator (57). Transcriptomic and expression experiments for *N. lanati* were conducted as part of this study (described later). These omics data sets were used to assign a confidence score to each gene in the model of *N. lanati*. Gaps in the model of *N. lanati* were filled by inspecting the EC assignments found for each other anaerobic fungus, as well as the GEMs of *E. coli* and *S. cerevisiae*, using the approach described above and looking for homologous genes in the genome of *N. lanati* (46, 58). The universal reactions and metabolites from the BiGG Models platform (59) was used to construct the *in silico* model where possible; if a reaction did not exist in that database it was manually added. The KEGG and MetaCyc databases were used as references to reconstruct the draft metabolic model based on the EC assignments of the metabolic annotation data (56, 60). The curated model for *N. lanati* was constructed by carefully following established genome-scale metabolic model construction protocols to refine the draft model (26). Specifically, each reaction was inspected to ensure consistency, mass, and charge balance where possible. Model quality was benchmarked by the Memote application (see data set S1 in the iNlan20 GitHub repository at the above-mentioned URL) (61). The curated *N. lanati* model as well as the entire reconstruction pipeline and all the data used in this work can be found in the model repository at https://github.com/stelmo/iNlan20. An experimentally measured flux of 1.5 mmol/g (dry weight)/h of glucose was used in all simulations. Flux balance analysis was used to simulate the genome-scale metabolic model of *N. lanati* using the COBRA Toolbox and COBRApy (62, 63). Flux samples ($N = 2,000$) were generated by sampling from the model and constraining the objective function to be within 90% of the optimum found by FBA. This threshold was set to reflect the assumption that the gut fungi need to maintain a high growth rate to compete with faster growing bacteria in their native microbiome (1). Escher was used to visualize the metabolism (64). Example code that can be used to run the model and computational experiments is supplied as an IPython notebook in the model repository available at the URL mentioned above.

**Culturing conditions used for experiments.** Standard anaerobic gut fungal culturing techniques were used (11) for all experiments. Briefly, *N. lanati* was grown at 39°C in sealed Hungate tubes (10-ml liquid volume) or 70-ml serum bottles (40-ml liquid volume) in both undefined complex medium C (MC) (65) and completely defined medium 2 (M2) (66), with 100% $CO_2$ headspace unless otherwise specified. Pressure accumulation in the headspace (67) was used as a proxy for growth, and the fungus was serially passaged after 2 to 3 days of growth. The carbon source was cellobiose (5 g/liter) unless otherwise noted. The cultures were not shaken.

**Genome and transcriptome isolation, sequencing, and analysis of *N. lanati*.** *N. lanati* was isolated from the feces of a sheep located at the Santa Barbara Zoo according to an established protocol (3). Fungal cell pellets for genomic DNA (gDNA) isolation were grown by inoculating 20 ml from a serum bottle of fungi in exponential phase (2 to 3 days of growth given a 10% inoculation volume into the

serum bottle) into a 1-liter bottle of medium C, using cellobiose as a carbon source. The serum bottle used to grow the inoculum was treated with chloramphenicol to reduce the risk of contamination. After 4 days of growth, the fungal cell mat was spun down and frozen at −80°C. Four of these frozen samples were subsequently shipped to the Arizona Genome Institute (University of Arizona, Tucson, AZ), where high-quality gDNA was isolated using a modified cetyltrimethylammonium bromide (CTAB) protocol (68). Briefly, these fungal cell mats were ground to a fine powder in a frozen mortar with liquid $N_2$ followed by very gentle extraction in CTAB buffer, which included proteinase K, polyvinylpyrrolidone, molecular weight 40,000 (PVP-40), and 2-mercaptoethanol (Sigma, St. Louis, MO), for 1 h at 50°C. After centrifugation, the supernatant was gently extracted twice with 24:1 chloroform–iso-amyl alcohol. The upper phase was removed, adjusted to one-tenth volume with 3 M potassium acetate, and gently mixed, and the gDNA was precipitated with iso-propanol. Subsequently, the gDNA was collected by centrifugation, washed with 70% ethanol, air dried for 20 min, and dissolved thoroughly in 1× Tris-EDTA (TE) buffer at room temperature. The purified gDNA was shipped to the JGI, where it was sequenced and annotated. Briefly, 10 μg of genomic DNA was sheared to approximately 15 to 20 kb using Megaruptor3 (Diagenode). The sheared DNA was treated with DNA Prep to remove single-stranded ends and then with DNA damage repair mix, followed by end repair, A tailing, and ligation of PacBio overhang adapters using SMRTbell template prep kit 1.0 (Pacific Biosciences). The final library was size selected with BluePippin (Sage Science) at a 10-kb cutoff size and purified with AMPure PB beads. PacBio Sequencing primer v3 was then annealed to the SMRTbell template library, and sequencing polymerase was bound to them using a Sequel binding kit 3.0. The prepared SMRTbell template libraries were then sequenced on a Pacific Biosystem's Sequel sequencer using 1M v3 SMRT cells, and version 3.0 sequencing chemistry with 10-h movie run times. Subsequently, the main assembly consisted of 62.05× of PacBio read coverage (7,821 bp average read size) and was assembled using MECAT version 1.8; the resulting sequence was polished using ARROW (version 2.2.3). The assembled genome was annotated using the JGI annotation pipeline. The genome is available at https://mycocosm.jgi.doe.gov/Neolan1.

RNA for transcriptome and expression analysis was isolated as previously described (3, 21), in the Biological Nanostructures Lab (University of California Santa Barbara, CA). For the transcriptome, the RNA was harvested from fungal cell pellets grown in serum bottles on a variety of substrates (cellobiose, filter paper, reed canary grass, and corn stover, solids loading 1% [wt/vol], in both medium C and medium 2) to capture as much transcript diversity as possible. For expression analysis, triplicate serum bottles of fungus grown on medium M2, using cellobiose as the sole carbon source, were used. The RNA was isolated and purified using an RNeasy kit (Qiagen, Germantown, MD). The concentration and quality of the RNA were measured on a Qubit (Qubit, New York, NY) and TapeStation 2200 (Agilent, Santa Clara, CA). The RNA used for the transcriptome was pooled in equal parts before sequencing. RNA libraries were made using NEBNext Ultra II directional RNA with mRNA purification beads (NEB, Ipswich, MA); these were subsequently sequenced on a NextSeq 500 (Illumina Inc., San Diego, CA) using high-output 300-cycle settings and 150-base pair paired-end reads (the resultant coverage is 470 and 364 for the transcriptome and expression analysis, respectively). The reads were assembled using Trinity (69). TransDecoder (https://github.com/TransDecoder/TransDecoder/wiki) was used to find the highest likelihood coding regions in the transcriptome. The transcript abundance was estimated using Kallisto (70). The raw assembled transcriptome and filtered output of TransDecoder may be found at https://github .com/stelmo/iNlan20. The mean expression data of transcripts mapped to genes may be found in the GitHub repository of the model available at the URL mentioned above.

A separate transcriptome was sequenced to aid the genome annotation; the same conditions as mentioned previously were used. Stranded cDNA libraries were generated using the Illumina TruSeq stranded mRNA library prep kit. mRNA was purified from 200 ng of total RNA using magnetic beads containing poly(T) oligonucleotides. mRNA was fragmented using divalent cations and high temperature. The fragmented RNA was reversed transcribed using random hexamers and SSII (Invitrogen) followed by second-strand synthesis. The fragmented cDNA was treated with end repair, A tailing, adapter ligation, and 10 cycles of PCR. The prepared libraries were quantified using KAPA Biosystem's next-generation sequencing library quantitative PCR (qPCR) kit and run on a Roche LightCycler 480 real-time PCR instrument. The libraries were then multiplexed into pools, and sequencing was performed on the Illumina NovaSeq 6000 sequencer using NovaSeq XP v1 reagent kits and S4 flow cell and according to a 2-by-150 indexed run protocol. This transcriptome may be found at https://mycocosm.jgi.doe.gov/Neolan1.

**High-performance liquid chromatography, gas chromatography, and liquid chromatography-mass spectrometry measurements.** Liquid samples for high-performance liquid chromatography (HPLC) analysis were stored in microcentrifuge tubes at −20°C for batch analysis. Sulfuric acid (0.5 M) was added to the samples (1 in 100 volumes), vortexed, and allowed to mix at room temperature for 5 min. Thereafter, the samples were centrifuged for 5 min at 21,000 × $g$ and filtered using a 0.22-μm syringe filter into HPLC vials. The samples were run on an Agilent 1260 Infinity (Agilent, Santa Clara, CA) using a Bio-Rad HPX-87H column (Bio-Rad, Hercules, CA). Samples were run at two column conditions to effectively separate all the fermentation products (71, 72). Succinate, lactate, cellobiose, glucose, and ethanol were measured at 50°C with a flow rate of 0.5 ml/min and run length of 30 min. Fumarate, formate, and acetate were measured at 25°C with a flow rate of 0.4 ml/min and a run length of 40 min. The mobile phase in both cases was 5 mM sulfuric acid, and the injection volume was 20 μl. Cellobiose, glucose, and ethanol were measured with a refractive index detector, and the other compounds were measured on a variable wavelength detector ($\lambda$ = 210 nm). Standard curves for each compound were made at 3 concentrations bracketing the range expected in the samples. Gas samples were analyzed on a Thermo Fisher Scientific TRACE gas chromatograph according to a previously established protocol (25). Standard curves of $H_2$ were made daily from supplier (Douglas Fluid & Integration Technology,

Prosperity, SC) mixed gas at 1, 2, and 5% (mole basis). Fatty acid composition and extracellular amino acid concentrations were measured on a liquid chromatograph-mass spectrometer (LC-MS) using an Agilent Polaris 3 $C_{18}$-ether 150-by-3.0 mm (part number [no.] A2021150X030) column. Total run time was 13 min and injection volume was 5 $\mu$l with a column temperature of 30°C and flow rate of 0.3 ml/min. The MS was run in negative ionization mode, with the gas temperature at 230°C and a flow rate of 12 liters/min. Standards were made to bracket the expected concentration ranges of the fatty acids.

**Biomass composition measurements.** The biomass composition of *N. lanati* was experimentally determined in triplicates for each major component (DNA, RNA, lipid, carbohydrate, and protein). Cultures grown in serum bottles on medium C, with cellobiose as the carbon source, were harvested during exponential phase for each measurement. DNA and RNA were immediately isolated from the wet cell pellet using a previously established CTAB protocol (73). A reference set of triplicate fungal mats was harvested at the same time and lyophilized to estimate the dry mass fraction of DNA and RNA. Lipids and total carbohydrates, which we assumed to be exclusively chitin, were isolated according to established protocols (74). Lipid composition was further refined by mass spectrometry using a fatty acid $C_{18}$-ether column as described above. Protein extraction from fungal cell pellets was according to a previously developed method (75), and the concentration was measured on a Qubit (Q33327; Thermo Fisher Scientific). The amino acid composition of the protein fraction was assumed to follow the amino acid distribution of the predicted proteome.

Additionally, both the growth and non-growth-associated maintenance (GAM and NGAM, respectively) functions were estimated from experimental data. Briefly, five triplicate sets of Hungate tubes with various concentrations of cellobiose (1, 2, 3, 4, and 5 g/liter initially) in medium M2 were inoculated with 1 ml each (total liquid volume, 10 ml) from a single serum bottle of *N. lanati* growing at exponential phase (3 days postinoculation) in medium M2 with cellobiose as the carbon source. Pressure accumulation was measured twice daily to calculate the fungal growth rate (67). Liquid and gas samples from each triplicate set were harvested during two time points in exponential phase, 24 h apart. The gas samples were analyzed on a GC to determine the $H_2$ fraction of the gas, and the liquid samples were analyzed by HPLC for organic acid concentration (see "High-performance liquid chromatography, gas chromatography, and liquid chromatography-mass spectrometry measurements" for details). After the last measurement, the fungal cell pellets were harvested by centrifugation, lyophilized, and weighed. The estimated growth rate for each sample was then used to extrapolate the dry cell mass at the respective time points (67). The fluxes of the fermentation products could then be estimated by the molar accumulation of each compound divided by the time between measurements and the difference in cell dry masses between these points. The difference in cell masses was taken because each mature cell lyses and dies; thus, its remaining biomass no longer contributes to metabolism. Finally, these estimated fluxes were used to constrain the model and maximize the ATP yield. The GAM and NGAM were then estimated by finding the line of best fit through the plot of maximum ATP yield predicted by the model and the growth rate associated with the fluxes previously measured (26).

**$^{13}$C metabolic flux analysis for *N. lanati*.** Three serially passaged Hungate tubes using [1,2-$^{13}$C]glucose as the sole carbon source in medium 2 at an initial concentration of 5 g/liter were used for the labeling experiment. Each Hungate tube was passaged during exponential growth phase, after which, the cell pellet and remaining media were frozen at −20°C for later processing. The medium was analyzed for glucose and fermentation products using the HPLC protocol described above. The pellets were lyophilized, after which GC-MS measurements were used to quantify the isotopic labeling of protein-bound amino acids, glycogen-bound glucose, and RNA-bound ribose as described previously (76). A carbon transition model for flux analysis was constructed using the genome-scale metabolic model of *N. lanati* as a basis for the flux reactions and biomass equation. Other carbon transition models were used to check the accuracy of the MFA model (77, 78). An ATP demand reaction was added to the carbon transition model and constrained to the maximum ATP flux capable of being generated by the system. This constraint ensured that the maximum feasible flux is routed through the hydrogenosome. INCA was used to perform the flux analysis and sensitivity calculations (79). The carbon transition model, constraints, and the GC-MS data can be found in the GitHub repository of the model available at https://github.com/stelmo/iNlan20.

**Model validation experiments.** Carbon utilization and vitamin essentiality experiments were conducted to test the predictive accuracy of the model. Carbon utilization was tested by growing *N. lanati* in medium 2 with each carbon source listed in Table 5 at 5 g/liter initial concentration instead of cellobiose. A carbon substrate was deemed able to support growth if the fungus could be passaged on it for 4 generations and still produce more than 8 lb/in$^2$ gauge of accumulated pressure (no-carbon blanks produce <1 lb/in$^2$ gauge of accumulated pressure). Similarly, the vitamin requirements of *N. lanati* were tested by individually removing each vitamin in medium 2 (listed in Table 5) and growing the fungus without it for 4 consecutive generations using cellobiose as the carbon source. Fluxes for comparing model predictions to experimental observations were measured similarly to how the fluxes for finding the GAM and NGAM functions were estimated; however, only 5 g/liter cellobiose loading was used. The total equivalent flux of glucose into the cell was calculated by measuring glucose accumulation and cellobiose depletion in the medium. It was assumed that *N. lanati* imports glucose, and not cellobiose, due to release of beta-glucosidases that decomposed the cellobiose in the medium.

**Hydrogenosome staining protocol.** The JC-1 dye was purchased from Invitrogen (part no. T3168; Carlsbad, CA, USA), and a standard protocol was used to visualize the presence of electrochemical gradients. Briefly, JC-1 was dissolved in dimethyl sulfoxide (DMSO; 1 mg/ml) and frozen until use. Dye aliquots were thawed and added to cultures of anaerobic gut fungal zoospores using final dye concentrations of 1 $\mu$g/ml. Zoospores were incubated with JC-1 for 30 min anaerobically in standard M2 medium

at 39°C. After incubation, cultures were filtered onto 3-$\mu$m polycarbonate membranes (part no. TSTP02500; Millipore Sigma, Burlington, MA, USA) with a nitrocellulose backing filter (part no. HAWP04700; Millipore Sigma). Cells were counterstained with 4′,6-diamidino-2-phenylindole (DAPI; 2 $\mu$g/ml) and mounted on glass slides using an antifade mounting solution composed of 4:1 Citifluor (part no. AF1; Electron Microscopy Sciences, Hatfield PA, USA) to Vectashield (part no. H-1000; Vector Laboratories, Burlingame, CA, USA). Prepared slides were placed on ice and imaged immediately using a Zeiss Axiovert M200 fluorescence micro-scope (Carl Zeiss AG, Oberkochen, Germany).

**Data availability.** The full genome, raw sequencing data, assembly, predicted genes, and annota-tions are available at https://mycocosm.jgi.doe.gov/Neolan1/Neolan1.info.html.

## SUPPLEMENTAL MATERIAL

Supplemental material is available online only.

**FIG S1**, TIF file, 0.2 MB.
**FIG S2**, TIF file, 0.1 MB.
**FIG S3**, TIF file, 0.1 MB.
**FIG S4**, TIF file, 0.7 MB.
**FIG S5**, TIF file, 0.1 MB.
**FIG S6**, TIF file, 0.1 MB.
**TABLE S1**, DOCX file, 0.1 MB.
**TABLE S2**, DOCX file, 0.1 MB.
**TABLE S3**, DOCX file, 0.1 MB.
**TABLE S4**, DOCX file, 0.1 MB.

## ACKNOWLEDGMENTS

This work was supported by funding from the National Science Foundation (NSF) (MCB-1553721). This work was part of the DOE Joint BioEnergy Institute (http://www.jbei.org) supported by the Office of Biological and Environmental Research of the DOE Office of Science through contract DE-AC02–05CH11231 between Lawrence Berkeley National Laboratory and the DOE. Research was sponsored by the Army Research Office and was accomplished under grant number W911NF-19-1-0010. The work conducted by the U.S. Department of Energy Joint Genome Institute, a DOE Office of Science user facility, is supported by the Office of Science of the U.S. Department of Energy under contract no. DE-AC02-05CH11231. S. E. Wilken received funding support from the Dow Discovery Fellowship.

The views and conclusions contained in this document are those of the authors and should not be interpreted as representing the official policies, either expressed or implied, of the Army Research Office or the U.S. Government. The U.S. Government is authorized to reproduce and distribute reprints for Government purposes notwithstanding any copyright notation herein.

We further acknowledge the use of the Biological Nanostructures Laboratory within the California NanoSystems Institute, supported by the University of California, Santa Barbara and the University of California, Office of the President. We thank Jennifer Smith for help in sequencing the transcriptome of the gut fungus. Use was made of computational facilities purchased with funds from the National Science Foundation (CNS-1725797) and administered by the Center for Scientific Computing (CSC). The CSC is supported by the California NanoSystems Institute and the Materials Research Science and Engineering Center (MRSEC; NSF DMR 1720256) at UC Santa Barbara. We also thank David A. Kudrna and Shanmugam Rajasekar at the Arizona Genome Institute for their help in isolating and purifying high-quality gut fungal gDNA. We thank Maciek R. Antoniewicz, at the University of Michigan, Ann Arbor, MI, for help in generating the measurement data for the $^{13}$C tracer experiments. We also acknowledge the use of the UC Santa Barbara NRI-MCDB microscopy facility and thank Weiwei Li in the Keller Lab at UC Santa Barbara for the LC-MS fatty acid and amino acid analysis.

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
