## [Reviewer comments · mSystems]

Experimentally validated reconstruction and analysis of a genome-scale metabolic model of an anaerobic *Neocallimastigomycota* fungus

St. Elmo Wilken, Jonathan Monk, Patrick Leggieri, Christopher Lawson, Thomas Lankiewicz, Susanna Seppala, Chris Daum, Jerry Jenkins, Anna Lipzen, Stephen Mondo, Kerrie Barry, Igor Grigoriev, John Henske, Michael Theodorou, Bernhard Palsson, Linda Petzold, and Michelle O'Malley

Corresponding Author(s): Michelle O'Malley, UC-Santa Barbara

Review Timeline:

Submission Date:

January 11, 2021

Accepted:

January 13, 2021

Editor: Stephen Lindemann

Reviewer(s): Disclosure of reviewer identity is with reference to reviewer comments included in decision letter(s). The following individuals involved in review of your submission have agreed to reveal their identity: Snorre Sulheim (Reviewer #1); Rajib Saha (Reviewer #2)

Transaction Report:

DOI: <https://doi.org/10.1128/mSystems.00002-21>

Response to reviewer comments

Original reviewer comments are in **black font**, replies are in **blue font**.

Please note that the figure and table numbers in the main text and supplement have been changed/removed to address reviewer comments. The model has also been updated to include more CAZyme genes, as well as a slightly more extended sulfur metabolism.

Reviewer 1

Summary

Wilken et al. have reconstructed a new genome-scale metabolic model of the anaerobic fungi *N. lanati*. This fungus plays an important role in the degradation of lignocellulose biomass in the digestive tracts of herbivores. As this is the first GEM for anaerobe fungi the model in itself is valuable as a template for reconstruction of similar species, taking into account the large overlap in annotated genes with related fungi. The authors use transcriptome data in the model reconstruction to identify high-confidence reactions in the model. In comparison with MFA (¹³C-data) the predicted fluxes are in good agreement. The authors focus on the hydrogenosome, a mitochondrion-like organelle that are involved in energy and hydrogen metabolism and make use of the model in a clever way to increase the understanding of the influence of different enzymes on the function of the hydrogenosome.

Major comments

1. Although the model reconstruction seems to be thorough and well described on the coarse-grained level, I think a more detailed description/changelog is necessary for the model reconstruction to be reproducible. Assuming certain

parts of the model reconstruction process were semi-automated using Matlab/python scripts I suggest that the authors to organize and document these scripts accurately and add them to the github repository. This should also include all the data necessary to perform the model reconstruction, e.g. any manual curation, the reversibility / change in Gibbs free energy that is pulled from metacyc/equilibrators. The github page should also host the memote report to make this data easily accessible. A snapshot of the github repository should be deposited to zenodo or similar to ensure future access to the data.

Thank you for suggesting this edit. We have included the entire reconstruction pipeline now in the Github repository. The readme of the repo guides the reader through the reconstruction process. The model was manually built using custom Julia and Python scripts, but these are wrapped in Jupyter notebooks (with comments) to make the reconstruction steps easy to understand. Please see <https://github.com/stelmo/iNlan20> for more details. Upon publication, we will create a snapshot of the repo on Zenodo as requested.

2. A discussion on the shortcomings or simplifications of the model is lacking in the discussion. Taking into account the large number of annotated genes (and also the size of other eukaryote GEMs) 589 genes a fairly small model not covering most of the >10 000 annotated genes (although not all of these encodes for enzymes).

Thank you for suggesting this. We have added a discussion section (Section 3 – page 20) to address these issues. We have also expanded the scope of the model to more comprehensively cover CAZyme metabolism. The model now covers a total of 1018 genes (of which 451 are associated with CAZymes). The high CAZyme fraction is indicative of the current state of

knowledge of gut fungi: their CAZymes are much better studied than their metabolism (e.g. identification and functional assignment). Despite this, of the 791 metabolic reactions in the model, only 216 do not have genes assigned to them (see the Memote report in the Github repo). Most of these are either transporters (83% of the transporters are unassigned) or involved in the lipid biosynthesis reactions, both of which are very understudied in the gut fungi. Please see the discussion section for more details.

3. The introduction and the analysis of the genome of *N. lanati* (and related organisms) focus on the CAZymes, but it is not clear from the manuscript to what extent this (obviously very important) metabolic capacity is covered by the model reconstruction. It would be useful to demonstrate that the model can grow using lignocellulose as a substrate to tie in this seemingly loose end.

Thank you for suggesting this edit. We have added this capability to the model and demonstrate such a simulation in the notebook file: “iNlan20 Example Simulations.ipynb” on Github (under the ReconstructionAndAnalysis directory). However, there are several issues with meaningfully incorporating lignocellulolytic CAZymes into a constraint-based model. The first issue is measuring the flux of the lignocellulosic breakout products into the fungal cells. Lignocellulose is insoluble and highly heterogenous, thus it is not clear what products are liberated from lignocellulose and metabolized (e.g. glucose, fructose, xylose, cellobiose, cellotriose etc.), in what order (e.g. do the CAZymes liberate the glucose based polymers first, do the fungi preferentially import glucose, etc.), and at what rate/flux (e.g. measuring the rate of cellulase action on lignocellulose is non-trivial (Bansal et al., 2009)). Second, it is highly likely that the gut fungi dynamically modulate the CAZymes they secrete, further

complicating any steady-state analysis (Haitjema et al., 2017a). Given these ambiguities, we opted to lump the CAZymes into two classes: cellulases and hemicellulases. We then introduced a generalized cellulose molecule that consists of cellobiose, as well as a hemicellulose molecule that consists of xylose. The external lumped cellulase and hemicellulase reactions then decompose these substrates into sugars that are then metabolized by the model. Given these caveats we primarily focus on using glucose, as part of a fully defined media formulation, for the majority of the simulations and analysis. The inclusion of these lumped CAZyme reactions has been noted in the manuscript.

4. This manuscript contains a large number of small tables. In my opinion the manuscript would improve greatly from summarizing these results in figures/figure panels and just include the tables in the supplement.

Thank you for suggesting this edit. We have removed Tables 7 – 10 from the manuscript. Table 10 (the flux variability analysis output) is now in the supplement (as Table S4) and Tables 7 – 9, as well as Figure S7, have been combined into a new figure (Figure 5) and removed from the manuscript.

5. Although I understand that the authors try to frame this work towards strain engineering I don't see that the section on strain engineering strategies brings much value to the table, and the authors should consider to leave this part out. Optionally, if the authors want to really explore how one can design a stable microbial cooperation they can run simulate co cultivations with relevant bacteria to understand which phenotypes that support a stable and optimal interaction.

Thank you for suggesting this edit. We have removed the strain design section as suggested and replaced it with a discussion centered around the model's shortcomings and possible uses to deepen the understanding of the

gut fungal metabolism. Please see the Discussion section (page 20) in the manuscript for more details.

6. Line 406: The measured growth rate indicates that the bifurcating hydrogenase is not used, although there may of course be other influencing factors as well. Along with the observed hydrogen production it seems like the results indicates that the bifurcate hydrogenase is not used, in contradiction with the title of this section. The authors should either change the section title or better explain why it is more likely that electron bifurcation is an active part of the hydrogenosome metabolism. How are the gene expression levels of relevant genes?

Thank you for asking for clarification on this point. We have changed the title of this section to reflect the remaining uncertainty about both the bifurcating hydrogenase and the proton pumping (ATP synthase) enzymes. The discussion section also highlights these uncertainties and suggests possible routes forward. However, to briefly answer the reviewer's question: the gene expression levels of the ATP synthase, and associated proton pumping modules, are low; while the expression levels of the putative bifurcating hydrogenase modules are relatively high. However, the bifurcating hydrogenase modules may be associated with other mechanisms (e.g. the ferredoxin-based hydrogenase), complicating a simple analysis, see Table 3 for more information. Recent high-resolution growth data suggests that the gut fungi modulate their growth rate over time, depending on the environmental conditions (Wilken et al., 2020). Unfortunately, high resolution H₂ production data is not available to directly test if H₂ production is low throughout the growth curve or if it peaks (this would be direct evidence of more PFO usage under certain growth regimes), as was found for the growth rate in the previously cited paper. We elaborate more on this

in the new discussion section (Section 3). In sum, proteomic and enzymatic characterization experiments focusing on isolated hydrogenosomal extracts will be necessary to answer the questions raised in this paper.

Minor comments

1. The title can be more specific describing the most important finding

Thank you for suggesting this edit. We believe that the most important result of this work is the introduction of the first GEM of an anaerobic gut fungus (this includes the high-quality genome and the other omics data), and its relevance as a template for GEM reconstructions of other gut fungi. Due to the understudied nature of the gut fungal metabolism we feel that a strong emphasis on any metabolic claims would be premature at this point (as discussed in Section 3). Instead, the model should serve as a foundation for the gut fungi community to build their own models using this one as a scaffold, and to identify high impact areas (e.g. the hydrogenosome) to be experimentally probed to improve the understanding of the gut fungal metabolism.

2. Use genome-scale metabolic models consistently (not genome-scale model).

Thank you for spotting this inconsistency. We have corrected this in the text wherever it appears.

3. Although there is a diversity of abbreviations for genome-scale metabolic models, GEM seems to be the one that is most often used and I encourage the authors to use this term instead of GSM.

Thank you for suggesting this edit. We have changed all GSM abbreviations to GEM.

4. Lytic polysaccharide monooxygenases (LPMOs) are of great interest because of the ability to increase hydrolysis of lignocellulose. Are any of the CAZymes LPMOs?

We did find very weak evidence of three genes (974317, 1314054, 1407525) that have putative LPMO activity, measured by homology to known AA9 family genes, see <http://www.cazy.org/AA9.html> and the dbCAN scan results on the github page (OmicsData\Cazymes\dbcann.out.txt). However, LPMOs are typically oxygen dependent for activity, and the gut fungi inhabit anaerobic environments (e.g. the rumen of large herbivores). Thus, it is unlikely that they make use of LPMOs to aid in the digestion of lignocellulose. It is more likely that the gut fungi have evolved other lignocellulose degrading machinery (e.g. the cellulosome, their vast repertoire of pectin/esterases/polysaccharide lyases etc.) as shown in Figure S1. For more details concerning their lignocellulolytic machinery we recommend these references in the manuscript (Haitjema et al., 2017b; Solomon et al., 2016).

5. Model naming: The common GEM nomenclature have been i-initials-number-of-genes (eg. iMK1208 for E. coli) or optionally, as for S. cerevisiae and for S. coelicolor, organism-name – GEM (e.g. Yeast-GEM). I suggest that the authors use either these approaches to name the model for consistency in the field.

Thank you for suggesting this edit. However, we have decided to keep the name “iNlan20” as the name of the model, as recently recommended in (Carey et al., 2020). This naming convention (i-species-year) seems more informative than either of the historical approaches.

6. Line 46: Is it the 1788 CAZymes in general or the 585 in the cellulosome that underscores the potential?

The gut fungi secrete both free floating (un-complexed), as well as complexed enzymes to degrade lignocellulose (Haitjema et al., 2017b; Henske et al., 2018; Solomon et al., 2016; Youssef et al., 2013). Their potential is highlighted by the sheer number as well as the diversity of CAZymes their genome encodes for, thus both numbers are of significance.

7. Line 53: What key aspects are clarified?

Thank you for suggesting this edit. The most important contribution regarding the hydrogenosome is (1) the presence of PFO (some earlier work suggested it was absent in the gut fungi) (2) the use of PFO vs PFL to produce H₂ (the only energetically feasible pathway to produce H₂ uses PFO, but many earlier papers suggested that PFL is used for this purpose, as explained in the text) (3) the possible presence of a bifurcating hydrogenase, proton pumping machinery and the ATP synthase. For the sake of brevity, we did not introduce these specific results in the abstract as they require some explanation and caveats (e.g. the putative nature of (3)).

8. Line 67-68: It is unclear what the authors mean with synthesizing multi-omic datasets.

Here we mean the combination of genomic and transcriptomic data to inform and expand on our understanding of gut fungal metabolism. However, GEMs can also use proteomic data to further refine metabolic models and thus are well suited to act as a scaffold for combining data to better understand an organism. However, we have changed “synthesize” into “combine” to aid in clarity in this sentence.

9. Line 74: The model provides a framework to guide strain engineering, not a framework for the actual genetic modification.

Thank you for suggesting this edit. We have modified this sentence to make this clear.

10.Line 132: add citation to papers describing the lack of growth of other fungi in defined media.

Thank you for suggesting this edit. We have added the necessary references.

11.Line 178: You mean gaps in the draft reconstruction?

Thank you for asking for clarification. Yes, we meant that gaps in the draft (pre-curation) model can be filled by inspecting the genomes of the other gut fungi. We have modified the text to indicate this.

12.Line 182: Considering that only half of the genes are annotated, I guess there might still be a significant difference between the strains that is yet to be known. Maybe secondary metabolism?

Thank you for asking for clarification. Yes, it seems likely that the secondary metabolism accounts for a large number of these unannotated genes. This is an area of active research in the gut fungal community. In Section 3 we now discuss this as a possible explanation for the large number of unannotated genes.

13.Line 195: What do you mean by "Based on the metabolic reconstruction of *N. lanati*"? Is this actually a draft reconstruction similar to the ones created by reconstruction pipelines? Or the list of enzymes annotated to the genome?

We are happy to clarify this point. We changed the sentence to indicate that the draft model was curated to produce the final model. We have also updated the methods section to indicate that the draft model was comprised of the manually identified metabolic reactions. The curation process (removing low evidence reactions, setting reaction directionality, adding annotations etc.) was performed on the draft model to yield iNlan20. This process is also clearly reflected in the reconstruction pipeline as shown in the Github repo of the model (see <https://github.com/stelmo/iNlan20>).

14.Line 228: The PPP does not only regenerate NADPH, but also provide nucleotide precursors (e.g. ribulose 5-phosphate). How are these building blocks produced when the PPP is incomplete?

Thank you for asking for clarification. Only glucose 6-phosphate dehydrogenase, 6-phosphogluconolactonase, phosphogluconate dehydrogenase and 6-phosphogluconate dehydratase are missing in the PPP pathway. The nucleotide precursor biosynthesis pathway is complete, i.e. 6-phosphogluconate dehydratase (PRPP) can be synthesized from glucose using the incomplete PPP pathway found in the gut fungi. In other organisms the aforementioned missing enzymes are primarily used to regenerate NADPH, but alternative pathways can also be utilized, as in the case of the gut fungi, see (Spaans et al., 2015) for other alternative pathways. We have added a sentence to this section making it clear that the nucleotide precursors can be synthesized.

15.Section starting on 247: Are there any other GEMs that describes the hydrogenosome? The discussion on the ATP synthase could benefit from an illustration.

Thank you for asking us to clarify this. To the best of our knowledge the hydrogenosome has not been incorporated in another GEM. The most studied organisms that are known to incorporate a hydrogenosome include: *Tritrichomonas foetus*, *Nyctotherus ovalis*, and *Trichomonas vaginalis*. Figure 3 serves as an illustration capturing the basic elements of all the known and putative pathways in the hydrogenosome. However, we have also included two references to the ATP synthase, generally and hydrogenosome specific respectively, to help illustrate this to the interested reader (Junge and Nelson, 2015; Muller et al., 2012).

16.Line 259: Is the ATP synthase essential?

Thank you for indicating that we should be clearer about the role of this enzyme complex in the gut fungal metabolism. First, the ATP synthase is not essential in the current model. We have changed the text to indicate that its flux is constrained to zero because the evidence for its function in the gut fungi is ambiguous. As noted in the main text, we could not find any of its membrane bound subunits, as previously observed, see (Seppälä et al., 2016) for more information. Without the membrane bound subunits this complex cannot function in its traditionally understood role. This then raises questions about the electrochemical gradient observed in Figure S4, which seems to indicate that a proton motive force is present in the hydrogenosome. We are not sure how to resolve this discrepancy with the current level of understanding of the gut fungal metabolism. We chose to be conservative in the model, only keeping energy generating pathways for which there is strong evidence. This is also discussed in Section 3.

17. Line 288: provide reference to other gut fungal species. Are these the 4 others shown in Figure 2?

Thank you for asking for clarification. Yes, Figure 2 contains information for *N. lanati* as well as 4 other gut fungi with publicly available genomes. Technically another genome is also available (*Piromyces sp. E2*), but the quality of its genome is very low and we opted to not include it for aesthetic purposes. However, the message of the figure is unchanged with or without *P. sp. E2*.

18. Line 310: Matthews coefficient of correlation is a good metric for these kind of model evaluations

Thank you for suggesting this edit. We have added the Matthews coefficient to describe the model's accuracy (it is 0.79) in the text as well as the table detailing the specific experimental tests.

19. Line 330: Was this constraint applied in FBA simulations as well?

Yes, the FBA model was constrained by independent experimental measurements as well. The text has been updated to make this clearer.

20. Line 334: Significance level? Explain the difference in the TCA-cycle use?

The fit of Figure S5 is significant ($P < 0.01$) and the text has been updated to make this clear. The discussion (Section 3) now attempts to explain the discrepancies observed in the TCA cycle. Briefly, we suspect that the proton pumping modules identified in the Table 3 may play larger metabolic role than currently modeled. However, without further experimental characterization this is hard to confirm.

21. Line 435: I think it should be noted here that unconstrained GEMs often have a large flux variability because the system of linear equations is usually (always) underdetermined.

We have modified this sentence to make it clear that this is typical of GEMs.

22. Figure S8: What is meant by "slack"?

Thank you for suggesting this edit. This figure has been removed.

23. Line 361: The footnote should be included in the main text.

The footnote has been inserted in the main text.

24. Line 364: Omit period in Figure 6.A (and similar occurrences).

All relevant figure references have been changed.

25. Line 364: Could the CO₂ production rate be used as well to determine the pathway usage?

Thank you for asking about this. Yes, ideally CO₂ measurements would shed some light on this since PFO produces CO₂ while PFL does not.

Unfortunately, the defined gut fungal media contains a bicarbonate buffer that regulates the pH of the media through CO₂. Additionally, we could not get the gut fungus to grow in a media that did not contain this buffer. Thus,

we could not use this route of analysis to shed more light on the hydrogenosomal metabolism. In Section 3 we do stress that further experimental work on the hydrogenosome would be greatly beneficial and this would likely involve addressing the CO₂ issue.

26.Line 397: replace ", see Table S3" with "(Table S3)". Same with similar occurrences.

Thank you for suggesting this edit. We have implemented this change across the manuscript.

27.Table 4 and 5: Move to Supplements and make summery figures in main text. Table 6-10 consider if the content is better conveyed through figure panels, with the current tables in the supplement.

Thank you for suggesting this edit. We have reduced the number of tables in the manuscript by combining these tables, moving some of them to the supplement and creating a new figure (Figure 5), which collates all their results.

28.What are the implications of the incomplete pathways for synthesis of folate, heme and biotin?

Thank you for asking about this. In defined media under the conditions we tested for, heme and 4-aminobenzoate (a folate precursor) were determined to be experimentally essential, as suggested by the model and reported in Table 4. However, experiments showed that biotin was not found to be a required component of the media. This suggest that the fungus does not need it or that the gene annotations are poor. We suspect that the latter is a better explanation. We have added text to this section to clarify this.

29.Line 132: Citations to growth data of other anaerobic fungi needed

Thank you for suggesting this edit. We have added references as required.

30.Line 251-253: Provide reference for this hypothesis.

Thank you for suggesting this edit. We have added references for this hypothesis.

31. Figure S4: Move methods description to M&M.

Thank you for suggesting this edit. We initially included the method used to stain the hydrogenosome in the supplement because it was not directly related to the model construction methods. However, it is now located in the Materials and Methods section under its own heading.

32. Line 525: What is the rationale behind this threshold

Thank you for asking about this. This threshold was chosen to reflect a qualitative need for the gut fungi to maintain their growth rate. In the rumen environment they must compete with faster growing bacteria, so we thought it was reasonable to constrain the lower bound of the growth rate to 90% of the optimum. In other work this threshold varies from 50% to 95% (Gudmundsson and Thiele, 2010; Mo et al., 2009; Park et al., 2012). We have added some justification of this in the text.

33. Seems like file names of supplemental files are used in both figure captions and supplemental txt. Correct this to the correct reference (e.g. correct "MFA model and data.xlsx" in the caption of Figure S5)

Thank you for suggesting this edit. We have removed all occurrences of this in the manuscript.

34. Remove track changes from Supplemental text.

Thank you for spotting this. We have removed the track changes.

35. Figure S7: It would be useful to show the ATP production by ATPase as a fraction of the total ATP production in the model.

Thank you for suggesting this edit. We have removed this figure but added this information to Figure 5.

36. ATP consumption constraint in Supp Material 2: Why isn't the ATP consumption dictated by the ATP maintenance requirements as determined in this work?

Thank you for asking for clarification. The MFA model does not account for maintenance requirements, hence the exclusion of the maintenance requirement found in the GEM model. However, the biomass objective function of the GEM was used to weight the precursor metabolites in the biomass equation for the MFA model. The metabolic network produces ATP, and thus a consumption term needed to be added. This ATP consumption reaction was set to the maximum value the model can produce. The reason for this constraint is due to the metabolic degeneracy associated with the hydrogenosomal and cytosolic substrate-level phosphorylation reactions. There is no carbon transition that can clearly separate the cytosolic flux through PFL from the hydrogenosomal PFL flux. The only difference between these two pathways is the amount of ATP generated (more using the hydrogenosome). Thus, we constrained the MFA model to produce as much ATP as possible, and thereby routing flux into the hydrogenosome. This assumption is qualitatively justified under biomass maximization that requires a high ATP production flux. We have added this caveat to the methods section dealing with the ^{13}C model.

Spelling errors

1. Line 50-51: Consider the use of *in vivo*, I suggest deleting it.

Thank you for suggesting this edit. We have omitted the use of “*in vivo*”.

2. Figure S4 caption: strain instead of stain.

Thank you for suggesting this edit. The caption has been changed to read “... selective stain...”.

Reviewer 2

Major Critiques:

1. The reviewer could not find the SBML file of the model. In addition, the reviewer could not find any mapping between BIGG reaction/compound ID's and KEGG/K-Base ID's. Since the COBRA community is moving toward a more generic/usable representation of a system, the reviewer strongly recommends the authors to do that.

Thank you for asking for clarification. We have added the entire model reconstruction pipeline, as well as the model, to a Github repository see “<https://github.com/stelmo/iNlan20>”. In the root directory of this repository the reviewer will find 2 model files: “iNlan20.json”, which is the COBRAPy format of the model, and “iNlan20.xml”, which is the SBML format version of the model. These models are identical except for the format difference, and both can be loaded in COBRAPy for analysis (we also supply an example analysis notebook under “ReconstructionAndAnalysis\iNlan20 Example Simulations.ipynb”). Note that both model files are duplicated in the “ReconstructionAndAnalysis” directory, so the notebook in this directory can simply be run without having to move files (the results of running this notebook are also viewable directly on Github). While the model cross-links reactions and compounds to a variety of databases (to varying levels of coverage, see the annotation field in the model), we have also extracted these mappings for convenience in a namespace mapping file saved in the Github repository of the model under

“ReconstructionAndAnalysis\suppdata\model_namespace_mappings.xlsx” for completeness. Note, we opted to use the BiGG namespace as our default namespace, in keeping with many other published GEMs (Kavvas et al., 2018; Monk et al., 2017; Seif et al., 2019). While not every metabolite and reaction has a BiGG identifier (e.g. the hydrogenosomal reactions and metabolites because they are not present in the database yet), the majority reactions and metabolites have cross-referenced annotations detailed in their annotation fields in the model (see the Memote report for a summary, located in the root directory of the Github repo of the model: “iNlan20_memote_report.html”).

2. Result and Discussion, Page 12, Lines 268-271: Authors claimed that no homologs of the membrane bound subunits of complex 1 were found and no explanation was provided for the activity of complex 2 without complex 1. Result and Discussion, 13, Lines 287-290: In lines 268-271, authors mentioned that no homologs of the membrane bound subunits of complex 1 were found. But here they mentioned, complex 1 is identified in *N. lanati* genome. It is a bit confusing.

Thank you for asking for clarification on these two points. Briefly, we find evidence for all the subunits of complex 2, but none of the membrane-bound subunits of complex 1 or the ATP synthase. This is in agreement with gut fungal literature on this topic (Seppälä et al., 2016). The reviewer is correct that without the membrane-bound subunits complex 1 and the ATP synthase cannot function as classically understood. We have modified the text to make this clearer and we have also added discussion section (Section 3) highlighting this gap in knowledge. Briefly, we provide two options. First, the missing membrane-bound subunit genes are present but unannotated, possibly because they are too divergent from known homologs. Second, the

membrane-bound subunits are missing, and we do not understand the new role these enzymes play in the metabolism. We note that further experimental characterization is required to answer this question. Finally, we included these components in the model to investigate the consequences of their addition in silico, and to see if the experimental observations can be reconciled with their addition, e.g. the bifurcating hydrogenase analysis. We have also added text to make this clear.

3. Result and Discussion, 14, Lines 310-313: The model did not show growth when xylose acts as sole carbon source, which is found experimentally. The authors mentioned about regulation or cellular imbalances as potential cause. Provided the tools and their own datasets, the authors should explore more and come up with a more solid explanation. Same is true for the rest of the inconsistencies.

Thank you for suggesting this edit. We have added a discussion section, as well as clarifications to the result sections to address the relevant inconsistencies mentioned by the reviewer. Throughout the manuscript we placed special emphasis on the hydrogenosomal metabolism, as noted in the other responses. Unfortunately, given the understudied nature of the gut fungal metabolism, the analysis usually raises more questions than it answers. This reflects the state of knowledge typical of non-model, complex organisms like the gut fungi and illustrates the need for a reconstruction that can begin to address these unknowns in a systematic fashion. Regarding the xylose issue, constraint-based models are good tools to use to understand cases where the model does not predict growth, but growth is observed experimentally (Feist et al., 2007; Mo et al., 2009; Orth and Palsson, 2012) (model false-negative prediction). Omics data can be used to find likely genes that would solve the issue (as was attempted for the membrane-bound

subunits of complex 1 and the ATP synthase). Unfortunately, the case for xylose is the converse situation (model false-positive prediction). This is more challenging as constraint-based models (at least as implemented here) cannot be used to interrogate regulation, environmental triggers etc. The most recent literature on this issue suggests that there might be a transporter issue (Henske et al., 2018), however we did find evidence of a xylose transporter in the genome. This suggests that cellular regulation might be a better explanation. This observation has been added to the manuscript.

4. Result and Discussion, 18, Lines 422-424: Previously authors mentioned that they artificially included ATP Synthase in the model but here authors are saying the impact of ATP Synthase in ATP production are very small. If this is the case, why ATP Synthase was included in the model?

Thank you for suggesting this edit. We have constrained the ATP synthase flux to zero in the base case model. The reviewer is correct that there is not enough evidence to justify its inclusion in the working model. However, the enzyme was not removed from the model completely, because of the perplexing staining results shown in Figure S4. Now the ATP synthase reaction needs to be manually un-constrained (as demonstrated in the example code, see “ReconstructionAndAnalysis/iNlan20 example simulations.ipynb” in the Github repository) to analyze its impact on the metabolism. We discuss the motivation for this in the manuscript, as well as in the new discussion section (Section 3). Briefly, a reduced proton pumping module has been observed in hydrogenosomes of other organisms (unrelated to the gut fungi) and the presence of the pH gradient (Figure S4), and the partial enzymatic machinery suggests that these enzymes might still play a role in the gut fungal metabolism. In the manuscript we suggest that

isolation and enzymatic characterization (assays and/or proteomics) of this organelle is required to address this lingering question.

5. **Materials and Method:** Providing gap filling information could have been useful especially it would have informed people working on other less characterized system. Did the authors use something like Gapfill or Optfill algorithm?

Thank you for asking about the model construction process. The entire model construction pipeline is now available on the Github repository. We did not use an algorithmic gap filling procedure during the model construction, instead we manually reconstructed the entire metabolism by comparing identified genes (based on the metabolic annotation data, see “MetabolicTables/Bidirectional annotation data.xlsx” in the Github repository of the model) to pathways found in KEGG and MetaCyc. When a step in a pathway was missing, we looked at the relative completeness of the rest of the pathway as well as experimental data before we added a reaction without genetic support, e.g. if one enzyme is missing in the biosynthesis pathway of an amino acid, and the gut fungus can grow in defined media lacking all amino acids, we gap filled that enzyme. This amounts to manually doing what Gapfill or Optfill do. The reason for the manual approach was to better understand the gut fungal metabolism through in-depth reconstruction. Furthermore, automated gap-filling tools require a set of reactions to pull options from. Due to the poorly studied nature of gut fungi a closely related organism to pull from did not exist. We worried that automated tools from distantly related organisms would add more incorrect content than it would add. The confidence associated with each reaction is based on the level of support for each reaction added, see Table S2, e.g. gaps

filled without any evidence received a score of 0. This makes it easy to improve on the model when more characterization is done.

6. Is there a reason why the authors choose to focus on the primary metabolism? Since they sequenced the genome and performed transcriptomics experiment, can they not make use of these datasets to look into the secondary metabolism? Have they decided to do that since they could only estimate fluxes in and around central metabolism?

Thank you for asking about this modeling simplification. We decided to focus on the primary metabolism because it is much better understood than the gut fungal secondary metabolism. We could have used the genome to look for secondary metabolite producing genes, but this area of the gut fungal metabolism is very understudied. For reference, the first genome was only sequenced in 2016 and gene annotations in the gut fungi present difficulties, e.g. highly repetitive sequences with low homology to other genes, see (Wilken et al., 2019) for a short review. The reviewer is also correct, it is much easier to measure fluxes of the primary metabolism, e.g. ^{13}C tracers are largely restricted to the central metabolism. Finally, it is also not clear how the secondary metabolites contribute to biomass formation, thus making their inclusion in the biomass objective function challenging. Thus, for a completely novel GEM we decided it would be better to only model the primary metabolism. We hope that our model will serve as a scaffold for further reconstructions focused on secondary metabolism. However, we have added a discussion centered around shortcomings found in the model in Section 3.

7. On model building, some of things are not clear such as the source of information on intracellular localization and (other than fatty acids) how the macromolecular compositions are broken down to their constituents. Which

method was used to incorporate the gene expression data into the model? Was it iMAT? If it was, what was the rationale behind this choice over GIMME or INIT? How many reactions can carry flux (not under biomass maximization but in general)? What about thermodynamically infeasible fluxes?

Thank you for asking for clarification. We followed the procedure detailed in (Thiele and Palsson, 2010) to estimate the components of the biomass function. As noted in the methods section, the DNA, RNA, lipids, carbohydrates and protein components were measured using established techniques as noted in the methods section (under “Biomass composition measurements”). Briefly, the DNA and RNA composition were assumed to follow those of the genome and transcriptome respectively, as sequenced for this work. The lipid fraction was approximated to follow the measured fatty acids, and the protein composition was assumed to match that of the predicted proteome. We assumed that the carbohydrate fraction was solely composed of chitin. The latter assumption was a simplification based on previous measurements that found that chitin was the major carbohydrate component of anaerobic gut fungi (Mountfort et al., 1994). For this work we only used gene expression data under cellobiose induced growth. The gene expression data was used to add confidence to the inclusion of reactions and genes in the model (see the notes section for each reaction in the model). The expression data was also used to investigate whether putative hydrogenosomal enzymes were expressed, as noted in Table 3. Given how understudied the metabolism of the gut fungi is (e.g. is there a bifurcating hydrogenase, is there a proton gradient that produces ATP?), we did not impose additional constraints using the gene expression data on the model (e.g. by using iMAT). There are 304 universally blocked reactions in the

model, see the Memote report in the root directory of the Github repo of this model. The model's reactions were manually curated using Equilibrator, MetaCyc and KEGG to ensure that there are no thermodynamically infeasible reactions. Additionally, the entire model construction pipeline is available on Github, see "ReconstructionAndAnalysis" in the root directory.

8. The R2 value of the linear plot (ATP vs. growth) is about 0.2. How could such a weaker correlation be used to calculate GAM and NGAM? As for the flux estimation, what method the authors use? Although it is not the major focus of the work, a bit more details will be helpful.

While the correlation of Figure S3 is weak as noted, the GAM and NGAM estimated here falls in the range of values typically used by other fungal GEMs, as shown in Table R1. Furthermore, it is often the case that groups fit both NGAM and GAM directly to match the measured growth rate of an organism. Indeed, this is the approach taken in (Tomàs-Gamisans et al., 2016; Vongsangnak et al., 2016). Given that the predicted and measured growth rate of *N. lanati* is close (0.044 vs. 0.045 ± 0.003 1/h), the estimated values likely wouldn't change much if we adopted the direct fitting approach (as opposed to indirectly fitting it using the current approach of Figure S3). As the reviewer notes, the GAM and NGAM estimates are not a major focus of this work, thus we believe that these estimates are sufficient. However, we have added references to the GAM and NGAM values used in other models to the caption of Figure S3 to highlight the uncertainty associated with our estimates.

Table R1: Assorted values of NGAM and GAM used in fungal GEMs.

Fungus	NGAM	GAM	Reference
N. lanati	2.27	76	This work
Mortierella alpina	1.9	71.4	(Ye et al., 2015)

Yarrowia lipolytica	7.9	86.8	(Pan and Hua, 2012)
Mucor circinelloides	1.9	72	(Vongsangnak et al., 2016)
Pichia pastoris	2.3	72	(Tomàs-Gamisans et al., 2016)
Saccharomyces cerevisiae	<1 – 18.7	15 – 91	(Villadsen et al., 2011)

For this work we estimated both NGAM and GAM as recommended in (Thiele and Palsson, 2010). We describe the measurement procedure in the methods section, see “Biomass composition measurements”, but briefly the measured fluxes for the GAM and NGAM calculation were estimated by $\frac{\Delta \text{ millimoles of } X \text{ produced over time } T}{(\Delta \text{ dried mass over time } T)(\text{time } T \text{ in hours})}$ (where X was: H₂, succinate, lactate, formate, ethanol, acetate and glucose). Given the non-model nature of the gut fungi it is difficult to reduce measurement variability to that seen in organisms like *E. coli*, thus the noisy fit seen in Figure S3. Reasons for this include that the gut fungi cannot be grown in chemostats and the inoculation technique is not as well controlled, see (Haitjema et al., 2014) for more details.

9. About PFO vs. PFL issue, the authors could easily calculate the energy cost involving each one of these. They could even perform a shadow price analysis.

Thank you for asking for clarification on this point. We have updated Section 2.5 to be more precise about what we mean by the energetic costs associated with each enzyme. In Figure 3 it can be seen that there is no energetic cost difference, from an ATP production point of view (now explicitly stated in the relevant section of the manuscript), between using PFL or PFO (Muller et al., 2012; van der Giezen et al., 1998). Each enzyme produces one ATP molecule for every pyruvate molecule imported into the hydrogenosome. Thus, from a constraint-based modeling perspective there will be no difference between these two enzymes. However, there may be a

protein synthesis cost difference between them. The genes for PFL encode for proteins with masses 89 and 90 kDa, while the PFO gene encodes for a protein with mass 539 kDa. Thus, there is a significant difference in energetic costs associated with the amino acid synthesis required to support the PFO pathway. However, there does seem to be a catalytic efficiency difference between these two enzymes, with PFO having a turnover rate 6 times higher than PFL. This might suggest that PFO is more efficient, which might be relevant under challenging culture conditions (e.g. when the fungus is grown on lignocellulose). This could partially explain the low expression of PFO in sugar rich environments, as used in this work. We have added a short discussion of this in Section 3 for completeness.

10. Figure 4: Although some of the estimated and model-predicted fluxes are close, there are so many fluxes from the TCA cycle and in hydrogenosome are different. The authors should discuss why this had happened or what could be missing. Figure 6 could be moved to supplemental file since it does not really say more than what was already mentioned in the main text.

Thank you for suggesting this edit. We have opted to remove Figure 6 and the ensuing strain engineering discussion and replaced it with a discussion section (Section 3) dealing with the shortcomings of the model, which includes why some of the flux predictions and measurements differ.

Minor Critiques:

1. Result and Discussion, Page 13, Lines 282-285: Homology sequences of bifurcating hydrogenase subunits should have been included in the Appendix.

Thank you for noting this omission. We have added the homology sequences to Github repository under “SupplementaryDatasets/Supplementary Data - 2

– Bifurcating hydrogenase hits in *N. lanati.fasta*’. The blast scores are noted in the readme of that same directory..

References

- Bansal, P., Hall, M., Realf, M.J., Lee, J.H., Bommarius, A.S., 2009. Modeling cellulase kinetics on lignocellulosic substrates. *Biotechnol. Adv.* 27, 833–848. doi:10.1016/j.biotechadv.2009.06.005
- Carey, M.A., Dräger, A., Beber, M.E., Papin, J.A., Yurkovich, J.T., 2020. Community standards to facilitate development and address challenges in metabolic modeling. *Mol. Syst. Biol.* 16, 1–9. doi:10.15252/msb.20199235
- Feist, A.M., Henry, C.S., Reed, J.L., Krummenacker, M., Joyce, A.R., Karp, P.D., Broadbelt, L.J., Hatzimanikatis, V., Palsson, B., 2007. A genome-scale metabolic reconstruction for *Escherichia coli* K-12 MG1655 that accounts for 1260 ORFs and thermodynamic information. *Mol. Syst. Biol.* 3, 1–18. doi:10.1038/msb4100155
- Gudmundsson, S., Thiele, I., 2010. Computationally efficient flux variability analysis. *BMC Bioinformatics* 11, 2–4. doi:10.1186/1471-2105-11-489
- Haitjema, C.H., Gilmore, S.P., Henske, J.K., Solomon, K. V., De Groot, R., Kuo, A., Mondo, S.J., Salamov, A.A., LaButti, K., Zhao, Z., Chiniquy, J., Barry, K., Brewer, H.M., Purvine, S.O., Wright, A.T., Hainaut, M., Boxma, B., Van Alen, T., Hackstein, J.H.P., Henrissat, B., Baker, S.E., Grigoriev, I. V., O’Malley, M.A., 2017a. A parts list for fungal cellulosomes revealed by comparative genomics. *Nat. Microbiol.* 2, 1–8. doi:10.1038/nmicrobiol.2017.87
- Haitjema, C.H., Gilmore, S.P., Henske, J.K., Solomon, K. V., De Groot, R., Kuo, A., Mondo, S.J., Salamov, A.A., LaButti, K., Zhao, Z., Chiniquy, J., Barry, K., Brewer, H.M., Purvine, S.O., Wright, A.T., Hainaut, M., Boxma, B., Van Alen, T., Hackstein, J.H.P., Henrissat, B., Baker, S.E., Grigoriev, I. V.,

O'Malley, M.A., 2017b. A parts list for fungal cellulosomes revealed by comparative genomics. *Nat. Microbiol.* 2, 1–8.

doi:10.1038/nmicrobiol.2017.87

Haitjema, C.H., Solomon, K. V., Henske, J.K., Theodorou, M.K., O'Malley, M.A., 2014. Anaerobic gut fungi: Advances in isolation, culture, and cellulolytic enzyme discovery for biofuel production. *Biotechnol. Bioeng.* 111, 1471–1482. doi:10.1002/bit.25264

doi:10.1002/bit.25264

Henske, J.K., Wilken, S.E., Solomon, K. V., Smallwood, C.R., Shutthanandan, V., Evans, J.E., Theodorou, M.K., O'Malley, M.A., 2018. Metabolic characterization of anaerobic fungi provides a path forward for bioprocessing of crude lignocellulose. *Biotechnol. Bioeng.* 115, 874–884.

doi:10.1002/bit.26515

Junge, W., Nelson, N., 2015. ATP synthase. *Annu. Rev. Biochem.* 84, 631–657.

doi:10.1146/annurev-biochem-060614-034124

Kavvas, E.S., Seif, Y., Yurkovich, J.T., Norsigian, C., Poudel, S., Greenwald, W.W., Ghatak, S., Palsson, B.O., Monk, J.M., 2018. Updated and standardized genome-scale reconstruction of *Mycobacterium tuberculosis* H37Rv, iEK1011, simulates flux states indicative of physiological conditions. *BMC Syst. Biol.* 12. doi:10.1186/s12918-018-0557-y

doi:10.1186/s12918-018-0557-y

Mo, M.L., Palsson, B., Herrgård, M.J., 2009. Connecting extracellular metabolomic measurements to intracellular flux states in yeast. *BMC Syst. Biol.* 3. doi:10.1186/1752-0509-3-37

doi:10.1186/1752-0509-3-37

Monk, J.M., Lloyd, C.J., Brunk, E., Mih, N., Sastry, A., King, Z., Takeuchi, R., Nomura, W., Zhang, Z., Mori, H., Feist, A.M., Palsson, B.O., 2017. iML1515, a knowledgebase that computes *Escherichia coli* traits. *Nat. Biotechnol.*

doi:10.1038/nbt.3956

- Mountfort, D., Orpin, C.G., Munn, E., Theodorou, M.K., Davies, D.R., Yarlett, N., Akin, D.E., France, J., Brownlee, A.G., Grenet, E., Fonty, G., Williams, A., Joblin, K.N., 1994. *Anaerobic Fungi: Biolog, Ecology, and Function*, First. ed.
- Muller, M., Mentel, M., van Hellemond, J.J., Henze, K., Woehle, C., Gould, S.B., Yu, R.-Y., van der Giezen, M., Tielens, A.G.M., Martin, W.F., 2012. Biochemistry and Evolution of Anaerobic Energy Metabolism in Eukaryotes. *Microbiol. Mol. Biol. Rev.* 76, 444–495. doi:10.1128/membr.05024-11
- Orth, J.D., Palsson, B., 2012. Gap-filling analysis of the iJO1366 *Escherichia coli* metabolic network reconstruction for discovery of metabolic functions. *BMC Syst. Biol.* 6, 30. doi:10.1186/1752-0509-6-30
- Pan, P., Hua, Q., 2012. Reconstruction and In Silico Analysis of Metabolic Network for an Oleaginous Yeast, *Yarrowia lipolytica*. *PLoS One* 7. doi:10.1371/journal.pone.0051535
- Park, J.M., Park, H.M., Kim, W.J., Kim, H.U., Kim, T.Y., Lee, S.Y., 2012. Flux variability scanning based on enforced objective flux for identifying gene amplification targets. *BMC Syst. Biol.* 6. doi:10.1186/1752-0509-6-106
- Seif, Y., Monk, J.M., Mih, N., Tsunemoto, H., Poudel, S., Zuniga, C., Broddrick, J., Zengler, K., Palsson, B.O., 2019. A computational knowledge-base elucidates the response of *Staphylococcus aureus* to different media types. *PLoS Comput. Biol.* 15. doi:10.1371/journal.pcbi.1006644
- Seppälä, S., Solomon, K. V., Gilmore, S.P., Henske, J.K., O'Malley, M.A., 2016. Mapping the membrane proteome of anaerobic gut fungi identifies a wealth of carbohydrate binding proteins and transporters. *Microb. Cell Fact.* 15, 212.

doi:10.1186/s12934-016-0611-7

Solomon, K. V, Haitjema, C.H., Henske, J.K., Gilmore, S.P., Borges-Rivera, D., Lipzen, A., Brewer, H.M., Purvine, S.O., Wright, A.T., Theodorou, M.K., Grigoriev, I. V, Regev, A., Thompson, D.A., O'Malley, M.A., 2016. Early-branching gut fungi possess a large, comprehensive array of biomass-degrading enzymes. *Science* (80-.). 351, 1192–1196.

doi:10.1126/science.aad1431

Spaans, S.K., Weusthuis, R.A., van der Oost, J., Kengen, S.W.M., 2015. NADPH-generating systems in bacteria and archaea. *Front. Microbiol.* 6, 742.

doi:10.3389/fmicb.2015.00742

Thiele, I., Palsson, B., 2010. A protocol for generating a high-quality genome-scale metabolic reconstruction. *Nat. Protoc.* 5, 93–121. doi:10.1038/nprot.2009.203

Tomàs-Gamisans, M., Ferrer, P., Albiol, J., 2016. Integration and Validation of the Genome-Scale Metabolic Models of *Pichia pastoris*: A Comprehensive Update of Protein Glycosylation Pathways, Lipid and Energy Metabolism. *PLoS One* 11, e0148031. doi:10.1371/journal.pone.0148031

van der Giezen, M., Kiel, J.A., Sjollem, K.A., Prins, R.A., 1998. The hydrogenosomal malic enzyme from the anaerobic fungus *neocallimastix frontalis* is targeted to mitochondria of the methylotrophic yeast *Hansenula polymorpha*. *Curr. Genet.* 33, 131–135.

Villadsen, J., Nielsen, J.C., Liden, G., 2011. *Bioreaction Engineering Principles*, 3rd Editio. ed. Springer.

Vongsangnak, W., Klanchui, A., Tawornsamretkit, I., Tatiyaborwornchai, W., Laoteng, K., Meechai, A., 2016. Genome-scale metabolic modeling of *Mucor*

circinelloides and comparative analysis with other oleaginous species. *Gene* 583, 121–129.

Wilken, S.E., Leggieri, P.A., Kerdman-Andrade, C., Reilly, M., Theodorou, M.K., O'Malley, M.A., 2020. An Arduino based automatic pressure evaluation system to quantify growth of non-model anaerobes in culture. *AIChE J.* 1–8. doi:10.1002/aic.16540

Wilken, S.E., Swift, C.L., Podolsky, I.A., Lankiewicz, T.S., Seppälä, S., O'Malley, M.A., 2019. Linking 'omics' to function unlocks the biotech potential of non-model fungi. *Curr. Opin. Syst. Biol.* doi:10.1016/j.coisb.2019.02.001

Ye, C., Xu, N., Chen, H., Chen, Y.Q., Chen, W., Liu, L., 2015. Reconstruction and analysis of a genome-scale metabolic model of the oleaginous fungus *Mortierella alpina*. *BMC Syst. Biol.* 9. doi:10.1186/s12918-014-0137-8

Youssef, N.H., Couger, M.B., Struchtemeyer, C.G., Liggenstoffer, A.S., Prade, R.A., Najar, F.Z., Atiyeh, H.K., Wilkins, M.R., Elshahed, M.S., 2013. The genome of the anaerobic fungus *Orpinomyces* sp. strain C1A reveals the unique evolutionary history of a remarkable plant biomass degrader. *Appl. Environ. Microbiol.* 79, 4620–34. doi:10.1128/AEM.00821-13

January 13, 2021

Prof. Michelle Ann O'Malley
UC-Santa Barbara
Chemical Engineering
MC 5080
Santa Barbara, CA 93105

Re: mSystems00002-21 (Experimentally validated reconstruction and analysis of a genome-scale metabolic model of an anaerobic *Neocallimastigomycota* fungus)

Dear Prof. Michelle Ann O'Malley:

Your manuscript has been accepted, and I am forwarding it to the ASM Journals Department for publication. Please note that Reviewer #1 has provided a short list of suggested further improvements to the text. For your reference, ASM Journals' address is given below. Before it can be scheduled for publication, your manuscript will be checked by the mSystems senior production editor, Ellie Ghatineh, to make sure that all elements meet the technical requirements for publication. She will contact you if anything needs to be revised before copyediting and production can begin. Otherwise, you will be notified when your proofs are ready to be viewed.

Sincerely,

Stephen Lindemann
Editor, mSystems

Journals Department
Supplementary Figure S5: Accept
Supplementary Figure S2: Accept
Supplementary Figure S1: Accept
Supplementary Table S1: Accept
Supplementary Table S2: Accept
Supplementary Figure S6: Accept
Supplementary Table S4: Accept
Supplementary Figure S3: Accept
Supplementary Figure S4: Accept
Supplementary Table S3: Accept

Review comments

Line 216-217: With the current text this seems contradicting. I guess it should be "*Of the 791 metabolic reactions...*". You could briefly note how many of these that are transport reactions which are in particular difficult to annotate and therefore quite often lacking gene assignment in GEMs.

Line 221-222: The added "*the genes associated with*" adds confusion rather than clarification to this sentence. The column heading also says "*Number of reactions that satisfy criterion*". I suggest to undo the change.

Line 592: Which threshold was used on Gibbs free energy change? I assumed you used these calculated values to determine the reversibility of the model. Was there contradicting information in metacyc vs equilibrator? If so, how was this handled? The methods section on model reconstruction could deserve a bit more detail, despite that the complete reconstruction is publicly available.

Line 612: Which flux sampling algorithm was used? Did the authors confirm that the random samples converged (see <https://www.nature.com/articles/s41540-019-0109-0>)?

Line 763: Can the authors define confidence intervals for the estimated values based on their line fitting?